# Identification and Characterization of Malate Dehydrogenases in Tomato (*Solanum lycopersicum* L.)

**DOI:** 10.3390/ijms231710028

**Published:** 2022-09-02

**Authors:** Muhammad Imran, Muhammad Zeeshan Munir, Sara Ialhi, Farhat Abbas, Muhammad Younus, Sajjad Ahmad, Muhmmad Kashif Naeem, Muhammad Waseem, Arshad Iqbal, Sanober Gul, Emilie Widemann, Sarfraz Shafiq

**Affiliations:** 1State Key Laboratory for Conservation and Utilization of Subtropical Agro-Bioresources, College of Agriculture, South China Agriculture University, Guangzhou 510642, China; 2School of Life Sciences, Tsinghua University, Beijing 100084, China; 3School of Environment and Energy, Peking University Shenzhen Graduate School, 2199 Lishui Rd., Shenzhen 518055, China; 4Department of Economics, Lahore College for Women University, Lahore 35200, Pakistan; 5Beijing Key Laboratory of Cardiometabolic Molecular Medicine, Institute of Molecular Medicine and Peking-Tsinghua Center for Life Sciences and PKU-IDG/McGovern Institute for Brain Research, Peking University, Beijing 100871, China; 6Department of Health and Biological Sciences, Abasyn University, Peshawar 25000, Pakistan; 7National Institute for Genomics and Advanced Biotechnology (NIGAB), National Agricultural Research Center (NARC), Park Road, Islamabad 45500, Pakistan; 8Center for Biotechnology and Microbiology, University of Swat, Mingora 19200, Pakistan; 9Department of Plant Breeding and Genetics, Ghazi University, Dera Ghazi Khan 32200, Pakistan; 10Institut de Biologie Moléculaire des Plantes, CNRS-Université de Strasbourg, 67084 Strasbourg, France; 11Department of Anatomy and Cell Biology, University of Western Ontario, 1151 Richmond St., London, ON N6A5B8, Canada

**Keywords:** malate dehydrogenase, genome analysis, QTL mapping, molecular docking, gene expression, salt stress, abiotic stress, tomato

## Abstract

Malate dehydrogenase, which facilitates the reversible conversion of malate to oxaloacetate, is essential for energy balance, plant growth, and cold and salt tolerance. However, the genome-wide study of the MDH family has not yet been carried out in tomato (*Solanum lycopersicum* L.). In this study, 12 *MDH* genes were identified from the *S. lycopersicum* genome and renamed according to their chromosomal location. The tomato *MDH* genes were split into five groups based on phylogenetic analysis and the genes that clustered together showed similar lengths, and structures, and conserved motifs in the encoded proteins. From the 12 tomato *MDH* genes on the chromosomes, three pairs of segmental duplication events involving four genes were found. Each pair of genes had a *Ka/Ks* ratio < 1, indicating that the *MDH* gene family of tomato was purified during evolution. Gene expression analysis exhibited that tomato *MDHs* were differentially expressed in different tissues, at various stages of fruit development, and differentially regulated in response to abiotic stresses. Molecular docking of four highly expressed MDHs revealed their substrate and co-factor specificity in the reversible conversion process of malate to oxaloacetate. Further, co-localization of tomato *MDH* genes with quantitative trait loci (QTL) of salt stress-related phenotypes revealed their broader functions in salt stress tolerance. This study lays the foundation for functional analysis of *MDH* genes and genetic improvement in tomato.

## 1. Introduction

Malate is an important tricarboxylic acid cycle intermediate product and a key molecule required for the cellular metabolism [1]. Malate dehydrogenase (MDH) is found in plants, animals, and microbes and catalyzes the reversible conversion of malate to oxaloacetate with the help of NAD+ or NADP+ as a cofactor, depending on the MDH isoform [2]. The chloroplast has NADP-dependent MDHs, whereas the cytosol, plastids, mitochondria, and peroxisomes contain NAD-dependent MDHs [3]. Plant MDH isoforms are usually encoded in the nucleus and exist as homodimers with comparable coenzyme binding sites, active sites, and quaternary structures. The MDH has two domains: an NAD/NADP-binding domain and a substrate-binding domain that are visually interlaced but have distinct functions. Furthermore, the active site of MDH proteins is located in a cleft between the two domains [4,5,6].

In higher plants, MDHs are classified according to their coenzyme specificity, physiological activities, and subcellular location. Cytoplasmic *MDHs*, for example, are engaged in both acid metabolism and carbon dioxide fixation in C4 plants, whereas mitochondrial *MDHs* are involved in the tricarboxylic acid (TCA) cycle [1,7,8,9]. The *MDH* gene family has been demonstrated to be important in a variety of anabolic and catabolic activities. Many *MDH* genes have already been characterized in a variety of species, including cotton [6,10,11] wheat [12], maize [8], apple [13,14], and *Arabidopsis* [15,16]. However, few studies have been conducted for MDHs at the genome wide level. For example, *Gossypium raimondi*, *Gossypium arboretum*, and *Gossypium hirsutum* each have 13, 13, and 25 MDH genes, respectively, and functional characterization studies showed that *G. hirsutum* cytosolic *MDH1* (*GhcMDH1*) plays an important role in fiber elongation [6,10,11]. In *Arabidopsis*, plastid-localized *NAD-MDH* play important role in embryo development, and energy homeostasis [15,16], while mitochondrial *MDH1* and *MDH2* control the seed maturation and post germination growth [17]. Furthermore, *MDH* genes were shown to be involved in plant responses to different abiotic stresses, e.g., cold and salinity [8,13,14]. In summary, *MDH* genes have undergone functional differentiation over the course of evolution in different species. It is thus essential to assess the evolutionary connections of *MDH* genes between different species.

Tomato (*Solanum lycopersicum* L.) is a model plant for studying fleshy fruits and highly significant vegetable crops grown worldwide [18]. The tomato genome is about 900 megabases (Mb), with approximately 35,000 predicted protein-coding genes [19,20,21,22]. Even though tomato production has been steadily increasing in recent years [18], abiotic stresses such as high temperatures, prolonged drought, and high salt concentrations have considerably reduced crop yield. Tomato plants are extremely sensitive to salt stress because excessive levels of Na+ ions disrupt cellular metabolism and ion balance. Identifying potential genes that regulate plant response to high salt concentration is critical for tomato molecular breeding. Thus, *MDH* genes present an excellent opportunity as MDH proteins were found to be the positive and negative regulators of plant responses to salt stress. For instance, rice plastidial NAD-dependent *MDH1* reduces the response to salt stress by modifying the vitamin B6 concentration of rice tissues [23]. The acquisition of salt tolerance by transgenic *Arabidopsis* was facilitated by the overexpression of plastidic *ZmNADP-MDH* in maize [8]. Likewise, apple cytosolic *MDH* can increase the cold and salt tolerance [13,14]. Recently, Chen et al. showed that the expression of the majority of poplar *MDH* genes was elevated under salt stress, indicating that *PtMDHs* play a crucial role in the salt tolerance mechanism [24]. However, there is little known about the *MDH* gene family in the genome of tomato and its expression pattern under salt treatment. Therefore, thorough characterization of *MDH* genes will improve our comprehension of their structure and role in the development of stress tolerant tomato crop.

In the present study, a comprehensive genome-wide analysis of tomato *MDHs* was performed. Overall, 12 *MDH* genes were identified in the tomato genome, and their chromosomal localization, physiological and biochemical properties, and evolutionary rate were estimated. Transcriptomic analysis of *MDH* genes in different tissues, various fruit developmental stages, and under abiotic stresses showed that tomato *MDHs* were differentially expressed in all these conditions. RT-qPCR result of *MDH* gene expression in response to salt stress, and the co-localization of *MDH* genes with salt stress related QTLs further confirmed their role in salinity stress tolerance. Molecular docking was also performed to identify MDH substrate preferences. In brief, our findings provide a fundamental understanding of the role of *MDHs* in regulating abiotic stress resistance in tomato and will be beneficial for long-term stress tolerance development in tomato.

## 2. Results

### 2.1. Identification and Characterization of MDHs in Tomato

The candidate *MDH* genes were identified through a systematic BLASTP search against the tomato genome database using amino acid sequences of MDHs from *Gossypium arboretum, Oryza sativa,* and *Arabidopsis thaliana* as queries. Pfam and InterProScan were used to validate MDH proteins, and a total of 12 *MDH* genes were found to be present in the genome of *S. lycopersicum* (Table 1). These tomato *MDH* genes were named as *SlMDH1* to *SlMDH12*, according to their chromosomal localization. In addition, the ProtParam tool (https://web.expasy.org/protam/; accessed on 25 January 2022) was utilized to characterize the physiological and biochemical parameters of the SlMDH proteins and the results are shown in Table 1. The number of amino acids contained in SlMDH proteins ranged from 298 to 467. The molecular weights of SlMDH proteins ranged from 31.81 to 51.66 kDa. Isoelectric points (*pI*) ranged from 5.35 to 8.9, with 7.14 serving as the mean value. Five of the SlMDH proteins were found to be localized in the chloroplasts, four in the cytoplasm, two in the mitochondria, and one in the glyoxysome.

### 2.2. Chromosomal Localization and Duplication Analysis of MDH Genes

To further study the genetic divergence and gene duplication, all *SlMDH* genes were mapped to their relevant chromosomes (Appendix A). The results showed that the 12 *SlMDH* genes were unevenly distributed on different tomato chromosomes. For example, chromosomes 1, 3, 8, and 9, each possessed 2 genes, while chromosomes 2, 7, 11, and 12 each had 1 gene. Genomic changes, including chromosomal rearrangements and gene duplication (Tandem and segmental duplication) occurrences are ascribed to the formation of the new gene family in the plant genome progression [25]. The gene duplication events were investigated for the *MDH* gene family in tomato and the results showed that there were two duplication events, i.e., one between *SlMDH4/SlMDH11* and the other between *SlMDH6/SlMDH12* (Appendix A). Interestingly, duplicated gene pairs were separated into different gene clusters, which suggests that the expansion of the *MDH* gene family in tomato was mainly attributed to segmental duplication events. The *Ka* (nonsynonymous)*/Ks* (synonymous) ratio is an important measure for determining the number of whole genome duplication events as well as the selection forces [26]. In general, a *Ka/Ks* >1 means positive selection, *Ka/Ks* < 1 indicates purifying selection and *Ka/Ks* = 1 stands for neutral selection [27]. The *Ka/Ks* ratio was estimated for the duplicated *SlMDH* gene pairs (*SlMDH4/SlMDH11* and *SlMDH6/SlMDH12)* and the results demonstrated that duplicated gene pairs had a *Ka/Ks* ratio of less than 0.2 (Appendix A). This lower *Ka/Ks* ratio suggests that tomato *MDH* experienced strong purifying selective pressure. 

To explore the evolution of *SlMDH* members, we performed comparative synteny and collinearity analysis on *MDH* genes of tomato, rice, and *Arabidopsis* (Figure 1). The syntenic map shows only two collinear gene pairs (*SlMDH9- LOC_Os10g33800.1* and *SlMDH1- LOC_Os04g46560.1*) between tomato and rice (Figure 1A). However, in the case of tomato and *Arabidopsis*, many collinear genes pairs were found (Figure 1B); *SlMDH7-AT4G17260.1*, *SlMDH6-AT1G53240.1*, *SlMDH6-AT3G15020.1*, *SlMDH9-AT1G04410.1*, *SlMDH9-AT5G43330.1*, *SlMDH5-AT3G47520.1*, *SlMDH3-AT2G22780.1*, *SlMDH2-AT2G22780.1*, *SlMDH2-AT5G09660.4*, *SlMDH12-AT1G53240.1*, and *SlMDH12-AT3G15020.1.* These results revealed common ancestors of these genes in rice, *Arabidopsis,* and tomato. In addition, there were also gene pairs with one, or two, *Arabidopsis* genes corresponding to the same or different tomato genes in the synteny blocks (*SlMDH9-AT1G04410.1/ AT5G43330.1*, *SlMDH12-AT1G53240.1/ AT3G15020.1*, *SlMDH6- AT3G15020.1/ AT1G53240.1*, and *SlMDH2-AT2G22780.1/ AT5G09660.4*). Such types of synteny events suggested that several *MDH* genes evolved before the divergence of tomato and *Arabidopsis* lineages.

### 2.3. Phylogenetic Analysis, Conserved Motifs, and Gene Structure Analysis of MDHs

To discover the evolutionary relationships between the MDH proteins of *Solanum lycopersicum*, *Arabidopsis thaliana*, *Oryza sativa*, *Gossypium arboretum,* and *Theobroma cacao* (Appendix A), a multiple sequence alignment was performed, and an unrooted phylogenetic tree was constructed based on alignment results (Figure 2). The results showed that 53 MDH sequences were clustered into five distinct groups. Of the 12 *SlMDH* genes, the groups I, II, III, IV, and V contained 2, 2, 2, 2, and 4 *SlMDHs*, respectively. Groups I, II, and III contained homologous genes with *Arabidopsis* and *Gossypium arboretum*, while group IV contained only one *Arabidopsis* homologous gene. Group V contained the most *MDH* members, with homologous genes of *Arabidopsis*, *Gossypium arboretum* as well as *Theobroma cacao*. In addition, it was found that the *Solanum lycopersicum*
*SlMDHs* were closely grouped with *Arabidopsis thaliana* and *Gossypium arboretum* within each group as compared to *Oryza sativa,* which may represent the evolutionary relationship between monocotyledons and dicotyledons and the conservation of MDH proteins during the evolution (Figure 2). Overall, the phylogenetic tree analysis revealed a highly conserved amino acid sequence, suggesting a strong evolutionary relationship within members of each subgroup of tomato MDHs.

*Solanum lycopersicum* MDH protein sequences and domains were analyzed using MEME to investigate the conserved motif locations (0–10) (Figure 3 left panel). Tomato MDH proteins have 5 to 10 conserved motifs based on their groups but motif 1, motif 7, and motif 9 are conserved among tomato MDHs. Motif 1 contains an active site (LTRLDHNRALGQI ) and is important for the catalytic functions of MDHs. Moreover, motifs that match the NAD-binding domain and the C-terminal domain of the MDH proteins were also present in all SlMDHs. In group IV (SlMDH7 and SlMDH8), motifs 2, 4, and 6 are missing, while motif 3 is absent in three members (SlMDH9, SlMDH4, and SlMDH11) of Group V. The results also showed that some motifs are only conserved in certain groups of MDHs. For example, motif 10 is only present in group V, while missing in group III (SlMDH5 and SlMDH10), group I (SlMDH6 and SlMDH12), group II (SlMDH2 and SlMDH3), and group IV. These results indicate the functional divergence of tomato MDHs.

To further understand the structural evolution of the *SlMDH* genes, we examined their gene structure (Figure 3, right panel). The number of introns in *SlMDH* genes ranged from 0 to 13. *SlMDH11* and *SlMDH4* exhibited the most introns, while *SlMDH5* and *SlMDH10* do not have introns. The numbers of introns and exons in *SlMDH* genes from the same group were similar, but there was a substantial variance between groups.

### 2.4. Putative Cis-elements in the Promoter Regions of MDHs

*Cis*-regulatory elements in the upstream (2 Kb) of the transcription start site of *SlMDH* genes were analyzed using the Plant Care database to explore the putative functions of *SlMDHs*. The results showed that *SlMDHs* have common TATA and CAAT boxes core *cis*-elements in their promoters (results not shown). Furthermore, among these highly conserved elements, mostly *cis*-elements were associated with important physiological processes, such as light sensitive, environmental stress related, development responsive, and phytohormones response elements (Figure 4). For example, hormone responsive element including ABA (Abscisic Acid), ERE (Ethylene responsive element), GA (Gibberellic Acid, ARE (Auxin responsive element), TCA-element and A-box; growth and developmental related *cis-elements*, such as O2-site, CAT-box and circadian *cis*-elements; and environmental stress related *cis*-elements, MBS, and LTR were found to be most abundant in promoters of *SlMDHs* genes. This implies that *SlMDHs* may have a role in developmental processes as well as environmental stress tolerance. Overall, the findings suggested that *SlMDHs* expression levels may fluctuate in response to phytohormones and abiotic stresses.

### 2.5. Expression Analysis of SlMDH Genes in Different Tissues and at Various Stages of Fruit Development

To gain more insights into the putative functions of tomato *MDHs*, the tissues specific expression profiles of *SlMDHs* genes were observed in ten diverse tissues and organs, i.e., roots, leaf, flower, fruits, and breaker fruit (a stage when the pink of tomato color first becomes noticeable) and mature green fruits, using the publicly accessible RNA-seq data from tomato. The results revealed that *SlMDH* genes were widely expressed at various plant developmental and in different tissues including flower bud, flower, fruit, root, and leaf (Figure 5), signifying the diverse biological functions of *SlMDH* genes. *SlMDH6*, and *SlMDH9* genes exhibit relatively the highest expression profiles among all the tissues. Similarly, *SlMDH11* also exhibited a higher expression level in all the tissues, except in the root. The *SlMDH2*, *SlMDH3*, *SlMDH10*, *SlMDH12,* and *SlMDH5* have almost medium expression levels among all the tissues compared to other genes. On the other hand, *SlMDH1*, *SlMDH4*, *SlMDH7,* and *SlMDH8* have variable gene expression depending on the tissues. For example, *SlMDH1* is mainly expressed in flower buds. Thus, it is noteworthy that *SlMDH* genes may play substantial roles in tomato developmental processes.

### 2.6. SlMDH Genes Expression in Response to Heat, Drought, and Salinity Stress

Using publicly accessible RNA-seq data, the expression patterns of putative *SlMDH* genes were explored from heat, drought, and salt stress under different timepoints (Figure 6). Of the 12 *SlMDHs*, 4 genes (*SlMDH2, SlMDH9, SlMDH6,* and *SlMDH11)* showed relatively higher expression level independent of stress response to all treatments indicating that the application of stress treatment did have a strong impact on the expression of these four genes. Moreover, among the four genes the expression level of *SlMDH2* decreased, while the expression level of *SlMDH6* increased along with the timepoint of salt treatment. However, *SlMDH3, SlMDH4, SlMDH10, SlMDH5,* and *SlMDH12* genes showed intermediate expression levels and they were expressed differentially under different stress treatment. Furthermore, some other genes were expressed at a lowest level depending on the stress, such as *SlMDH8* that was expressed at lower levels in response to heat stress and drought stress. The *SlMDH1* and *SlMDH7* genes were expressed at relatively low levels including in control and in response to salt, drought, and heat stresses, indicating that these genes have least role to alleviate the different stresses. Overall, among the four highly expressed genes (*SlMDH2, SlMDH9, SlMDH6,* and *SlMDH11),* only *SlMDH2* showed a clear pattern of decrease in the expression in the drought and salt stress conditions, while it showed an increase in the expression in the recovery phase (Figure 6).

### 2.7. MDH Quantitative Expression under Salinity Stress

To validate the RNA-Seq data, we further performed qRT-PCR analysis on plants exposed to salt stress (Figure 7). In response to salt stress, gene expression was examined at 0, 3, and 6 hours after treatment. The results showed that *SlMDHs* present differential expression in response to salt stress. For example, *SlMDH5*, *SlMDH7,* and *SlMDH8* gene expression was increased at 3 h and decreased at 6 h as compared to the control, while *SlMDH3* and *SlMDH10* showed an opposite trend at the same time points, respectively. Similarly, *SlMDH2*, and *SlMDH4* gene expression was decreased upon salt treatment, while *SlMDH12* expression increased progressively over time as compared to the control. Moreover, the expression level of *SlMDH1*, and *SlMDH11* decreased at 3 h, while slightly increased at 6 h but less as compared to the control. Similarly, *SlMDH6* and *SlMDH9* did not show any difference at 3 h and 6 h as compared to the control.

### 2.8. Molecular Docking of SlMDH

To predict the binding preference of SlMDH proteins with their substrates (malate and oxaloacetate) and cofactors (NAD+ and NADP), molecular docking was performed. Because *SlMDH2, SlMDH6, SlMDH9,* and *SlMDH11* are generally highly expressed among tomato *MDHs*, therefore we selected these four candidates for molecular docking. The binding affinity of each ligand to the SlMDH protein is tabulated in Table 2. 

The lowest binding energy in kcal/mol implies the best intermolecular binding conformation and vice versa. The substrates and cofactors interacting residues are shown in Figure 8 and Table 2.

SlMDH2, SlMDH6, SlMDH9, and SlMDH11 showed differences in their binding affinity with malate, oxaloacetate, NAD+ and NADP. For example, the binding affinity of SlMDH2 (−4.7) and SlMDH9 (−4.7) for oxaloacetate was higher than that of SlMDH6 (−4.6) and SlMDH11 (−3.7). Similarly, the binding affinity of SlMDH2 (−4.4) for malate was higher as compared to SlMDH9 (−4.2), SlMDH6 (−4.1), and SlMDH11 (−3.7). In general, all the proteins (SlMDH2, SlMDH6, SlMDH9, and SlMDH11) showed higher affinity for the cofactors as compared to substrates. For example, SlMDH6 showed the highest affinity for NADH (−10) and NAD+ (−9.9) as compared to malate (−4.1) and oxaloacetate (−4.6). However, the NADH molecule showed the strongest binding with SlMDH6, SlMDH9, and SlMDH11 among the tested substrates and revealed several strong hydrophilic and hydrophobic interactions. Furthermore, SlMDH2 showed a similar binding affinity between NAD+ and NADH.

SlMDH2, SlMDH6, SlMDH9, and SlMDH11 showed differences in their hydrogen bonding with their substrates and cofactors. For example, SlMDH6 formed a total of 25 hydrogen bonds with substrates and cofactors, while SlMDH9, SlMDH11, and SlMDH2 formed a total of 16, 11, and 11 hydrogen bonds, respectively. In addition, the number of hydrogens bonds formed between malate, oxaloacetate, NAD+ and NADH also varies among SlMDH2, SlMDH6, SlMDH9, and SlMDH11. For example, SlMDH6 showed 10 hydrogen bonds with NAD+, while SlMDH11, SlMDH2 and SlMDH9 showed 2, 4, and 3 hydrogen bonds, respectively. Similarly, SlMDH6 showed 6 hydrogen bonds with oxaloacetate, while SlMDH11, SlMDH2, and SlMDH9 showed 3, 2, and 5 hydrogen bonds, respectively.

In addition to binding affinity and number of hydrogen bonds, SlMDH2, SlMDH6, SlMDH9, and SlMDH11 also differed in their interacting residues. For example, oxaloacetate interacted with SlMDH11 through only one chain of residues Asp198, Cys418, and His421 (one hydrogen bond each), whereas oxaloacetate interacted with SlMDH6 through only one chain of residues Gly232, Gln192, Asp160, and His188 (one hydrogen bond each) and Arg163 (two hydrogen bonds). Together, our docking results highlighted the differences between SlMDH2, SlMDH6, SlMDH9, and SlMDH11 for their substrate and cofactor affinity.

### 2.9. Co-Localization of MDH Genes with the Salt Stress-Related QTLs

To gain a better understanding of the function of *MDHs* in the salt tolerance mechanism, *SlMDH* genes were found to be co-localized with QTLs associated with salt tolerance that had previously been described by [28,29,30,31,32]. These QTLs were identified and reported in the past based on morpho-biochemical characteristics. These characteristics include the following: time to flower (Flw3.1/FlW9.1), time to ripe (RIP3.1/RIP3.2), leaf length (Leaf9.1/Leaf9.2), leaf area (LA), dry shoot weight (DSW), number of fruits ripen (NFR), fruit weight (FW1.1), fruit firmness (Firm11.1/Firm11.2), soluble solid content (SSC11.1/SSC12.1), Na+ concentration in leaves (lnC9.1), and K+ concentration in leaves (lkn1.1). On chromosome 1, 13 salt related QTLs were identified from the previous studies [28,29]. Among these, SSC1.2 QTL is co-localized with the *SIMDH1* gene (Figure 9). Chromosome 2 has *SIMDH3*, which is co-localized with GH3.3 [32]. Chromosome 3 had 2 *SIMDH* genes (*SlMDH4* and *SlMDH5*), but only *SIMDH4* is co-localized with the flw3.1 and fw3.1 QTLs [31]. *SIMDH8* gene on chromosome 8 is co-localized with previously reported lnC9.1 QTLs [31]. Both chromosomes 11 and 12 had one gene (*SIMDH11*, *SIMDH12*), which were co-localized with the Firm11.1 and RIP12.1 QTLs, respectively [29]. *SlMDHs* co-localize with salt-related quantitative trait loci (QTLs), indicating that they play a role in a range of morpho-biochemical features that are influenced by salt stress.

## 3. Discussion

Malate dehydrogenases mediate the reversible conversion of malate to oxaloacetate and play critical roles in energy balance, plant growth, and cold and salt tolerance. MDHs are very active in plant cell and is required for numerous metabolic activities [14]. Numerous investigations have revealed that plant *MDHs* genes play an essential role in reacting to abiotic stress, such as low temperature [14], salt [8], and aluminum [33]. Until now, the majority of *MDHs* genes research has been conducted on crops such as cotton [6], corn [34], wheat [12], and fruits, such as apple [14] and grape [35]. However, the *MDH* gene family under salt stress has not been investigated yet.

In this study, we revealed that the tomato genome has 12 *MDH* members, which is less than the *MDH* numbers identified in *Gossypium arboreum* [10] and *Gossypium hirsutum* [6]. Based on phylogenetic analysis, the SlMDHs members were grouped with the *MDH* gene from *Arabidopsis*, indicating a close relationship between tomato and *Arabidopsis*. Furthermore, a total of 10 motifs were identified in the amino acid sequence of SlMDHs. However, even though the sequences in the various groups contain motifs of varying sorts and quantities, all of the sequences contain motif 1 (active site) as well as other motifs that match the NAD-binding domain and the C-terminal domain of the MDH proteins. A prior study found that the crucial active site residue differentiates malate dehydrogenase from lactate dehydrogenase, two enzymes with significant sequence similarity [36]. The five groups of *SlMDH* genes each contain a unique structure of introns and exons and a varied total number of introns. Unlike the other groups and consistent with cotton *MDHs* [6], the genes in group I have no introns. According to Ren et al., when compared to genes with lower expression levels, highly expressed genes have more and longer introns, implying that the *SlMDH* genes in group I are expressed at low levels in response to biotic or abiotic stressors [37].

The expansion of gene families is generally accomplished by the process of gene duplication, which essentially encompasses the processes of tandem duplication, segmental duplication, and transposition [38]. Among the 12 *SlMDH* genes distributed unevenly on eight chromosomes, two pairs of *SlMDH4/SlMDH11* and *SlMDH6/SlMDH12* of segmental duplication events were identified. *SlMDHs* genes have light-sensitive, environmental stress-related, development-responsive, and phytohormone response elements in their promoters, in addition to TATA and CAAT box core cis-elements, suggesting that *SlMDHs* play a key role in responding to developmental and environmental stimuli. Surprisingly, the duplicated genes had distinct cis-elements in their promoters, implying functional divergence in response to certain stimuli. According to an examination of the data from the transcriptome, the expression patterns of different *SlMDH* genes under salt stress are distinct from one another. This suggests that the roles of some *SlMDH* genes may have changed during the evolution of the gene family.

*SlMDH* genes were differentially expressed based on their transcriptome data, where *SlMDH9, SlMDH2, SlMDH6,* and *SlMDH11* genes showed similar pattern of higher expression, and other members exhibited a distinct expression in all tissues, respectively. The transcription levels of highly expressed *SlMDH* genes in roots subjected to salt stress were determined using qRT-PCR. The expression of the vast majority of *SlMDH* genes saw a considerable rise in comparison to the control, especially the *SlMDH5, SlMDH7,* and *SlMDH8* were highly expressed at 3 h, while, *SlMDH3, SlMDH10,* and *SlMDH12* at 6 h, respectively, indicating that these genes play an important role in early salt stress response, and may serve as candidate genes for future molecular mechanisms research. Among the 12 *SlMDHs*, *SlMDH2* is the only one to be localized in glyoxysome, which contains the enzymes of fatty acid oxidation and the glyoxylate pathway [39]. Among the five *MDHs* that are predicted to be localized in chloroplast in tomato (*SlMDH3*, *SlMDH4*, *SlMDH5*, *SlMDH8,* and *SlMDH11*), *SlMDH8* and *SlMDH3* showed the highest expression in response to early salt stress. Thus, in reaction to unfavorable environmental pressures, chloroplasts communicate their status with the nucleus through a process called retrograde signaling, which helps regulate the nuclear stress response [40]. Consistent with this observation, and an early increase in *SlMDH8* and *SlMDH3* expression in response to salt stress, it appears that both of these *SlMDHs* may be involved in early salt stress response in tomato. Indeed, we also found that *SlMDH8* and *SlMDH3* were co-localized with salt stress related QTLs. However, further studies are required to validate whether and how *SlMDH3* and *SlMDH8* are involved in salt stress tolerance in tomato.

## 4. Materials and Methods

### 4.1. Identification of MDH Proteins

*Solanum lycopersicon* MDH family sequences were retrieved from Phytozome v12.1 [41]. *Arabidopsis thaliana*, *Gossypium arboretum,* and *Oryza sativa* amnio acid sequences (Appendix A), obtained from the Phytozome v12.1 database, were used as queries in BLASTP [42] searches against *Solanum Lycopersicon* genome database. Then, all of the identified MDH proteins were subjected to the InterProScan (http://www.ebi.ac.uk/interpro/search/sequence-search; accessed on 25 January 2022) to confirm the presence of the MDH domain [43]. The SMART and Pfam databases were used to analyze each member of the MDH gene family [44]. Finally, the physicochemical parameters of the full-length tomato MDH proteins were calculated by ProtParam tool (https://web.expasy.org/protam/; accessed on 25 January 2022) [45]. The subcellular localization of each MDH protein was predicted using WoLF PSORT (http://www.genscript.com/wolf-psort.html; accessed on 25 January 2022) [46]. 

### 4.2. Synteny and Collinearity Analysis

*SlMDH* genes were displayed on chromosomes using Tbtools [47], based on the genome annotation file of *S. Lycopersicon*. A genome blast was run in Tbtools using the Quick Run MCScanX Wrapper option to conduct a synteny analysis. The results of this run were then visualized using the same program. The collinearity analysis was performed in Tbtools via selecting One Step MCScanX. In addition, the *Ka/Ks* values were derived with the assistance of Tbtools [27].

### 4.3. Sequence Alignment and Phylogenetic Analysis

Multiple sequence alignment of SlMDH proteins has been performed with the ClustalW program with standard setting [48]. The neighbor-joining (NJ) with the Poisson model was used to construct phylogenetic trees with a bootstrap value of 1000 replicates in MEGA 6. 

### 4.4. Structure and Conserved Motif Analysis of MDHs

Exon introns structure of *SlMDH* genes were displayed using Gene Structure Display Server 2.0 [49]. The conserved motifs of the SlMDH were located using the MEME server. The motifs were selected as 10 and the other parameters were used as the default [50].

### 4.5. Promoter Analysis of SlMDH Genes

The upstream 2000 bp of DNA sequence from the transcription start site of each *SlMDH* gene was derived from Phytozome database v12.1. After that, the sequences were analyzed for a *cis*-element using the PlantCARE database [51]. The *cis*-elements present in the promoter sequences of *SlMDH* genes were visualized using Tbtools [47]. 

### 4.6. Transcriptomic Data Analysis of MDH Genes

The expression data for *SlMDH* genes (bud, flower, leaf, and root) and six fruit developmental stages (young fruits approximately 1 cm in diameter at 2 weeks after pollination) 1 cm fruit, 2 cm fruit, 3 cm fruit, mature green (mature green fruits at 7 weeks after pollination), breaker fruits (when the color of mature fruits changes from green to faint yellow-orange), and breaker after 10 days (Br + 10) were retrieved from the Tomato Functional Genomics Database (TFGD, http://ted.bti.cornell.edu/; accessed on 25 January 2022) [52]. The Illumina high throughput RNA-Sequencing data of tomato under heat and drought conditions (GEO accession: GSE151277), as well as salinity (GEO accession: GSE148353), were obtained from the NCBI GEO database. The fragments-per-kilobase-per-million (FPKM) method was followed to analyze the expression. The TBtool was used to create the heatmap charts [47]. 

### 4.7. Plant Growth and Treatments

The tomato (*Solanum lycopersicum* L. cv. Rio Grande) seeds were sown and plants were grown at a temperature of 25 °C by day and 22 °C by night, with a humidity level of 60%, 12,000 lx of light, and a light/dark cycle of 16/8 h in a growth chamber. One week after germination, seedlings of the same length were transferred to a 1/2 Hoagland solution with a pH of 5.0 as reported earlier under the same growing conditions [53]. The Hoagland nutrient solution contained 1 mM (NH_4_)_2_SO_4_, 1 mM KH_2_PO_4_, 1 mM Ca(NO_3_)_2_·4H_2_O, 1 mM Mg_S_O_4_·7H_2_O, 2 mM Na_2_SiO_3_·_9_H_2_O, 20 μM Fe-EDTA, 1 μM ZnSO_4_·7H_2_O, 9.1 μM MnSO_4_,0.1 μM CuSO_4_·5H_2_O, and 10 μM H_3_BO_3_. After two weeks, tomato seedlings were subjected to high salinity treatment (250 mM NaCl solution) for 0, 3, and 6 h, while the control plants were grown under half-strength Hoagland’s solution without salt stress during this time. All the collected roots samples were frozen in liquid nitrogen and stored at −80 °C.

### 4.8. RNA Isolation, and qRT-PCR Analysis

The total RNA from was extracted from roots that were harvested at 0, 3, and 6 h of salinity stress using the RNA Plant Mini Kit (Tiangen, China). With the use of a Prime ScriptTM RT kit, a total of 2 g of RNA was transformed into cDNA (Takara, Shuzo, Otsu, Japan). qRT–qPCR with SYBR Green I Master Mix was performed using LightCycler^®^ 480 System (Roche, Germany) using gene specific primers (Appendix A) as described previously in [54]. The 2^−∆∆CT^ method was used to calculate the relative expression level of each gene, and the data were normalized using the Actin level [55]. The SPSS 11.5 package for Windows (SPSS, Inc., Chicago, IL, USA) was used for statistical analysis in this work. The Student’s *t*-test was used to examine the differences between the two groups of data. Results with a corresponding probability value of * *p* < 0.05 and ** *p* < 0.01 were considered to be statistically significant, respectively.

### 4.9. MDH Genes Localization with Salt Tolerance Related QTLs

Salt tolerance related QTLs for morphological and biochemical traits were retrieved from the Sol Genomics website, and past year’s publications [28,29,30,31,32]. The linked molecular markers of respective QTLs were also obtained from the tomato marker database and previous year’s publications [28,29,30,31,32]. To obtain the physical position of each marker, the marker sequence or name was BLAST against the tomato genome using Sol Genomics Network and the tomato marker database. *MDH* genes co-localized with salt tolerance related QTLs were displayed using mapchart software [56]. This demonstrated the MDH gene distribution along the surrounding QTLs. Genes and co-localized QTLs are visualized in red color.

### 4.10. Molecular Docking

Molecular docking was performed using a blind docking approach through AutoDock Vina [57]. The predicted protein models and the ligands; Oxaloacetate, NAD+, NADH and malate were minimized and converted to .pdbqt in PyRx 0.8 [58]. The ligand structures were drawn in ChemDraw 12.0 [59] and minimized in UCSF Chimera 1.15 [60]. The docking calculations were performed for 100 iterations in the case of each ligand and the one with the lowest binding energy was ranked the stable binder. Further, the Prank web (https://prankweb.cz/ accessed on 25 January 2022) server was used to predict the binding residues of each enzyme.

## Figures and Tables

**Figure 1 ijms-23-10028-f001:**
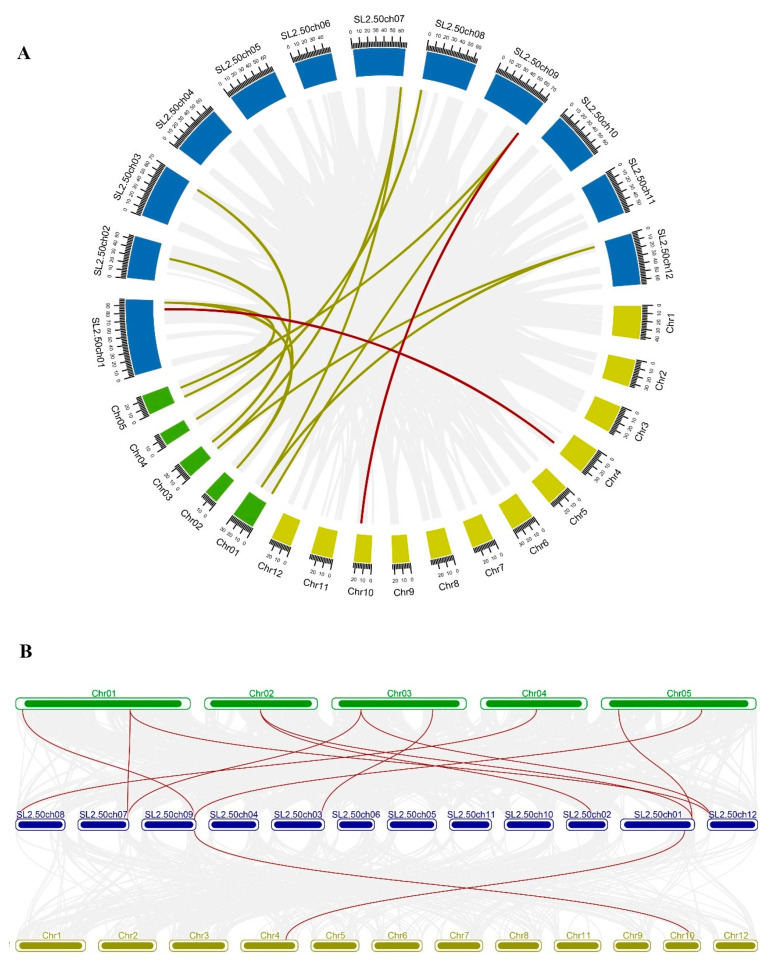
(**A**) Circos plot of *MDH* genes in the tomato, *Arabidopsis*, and rice genome. (**B**) Collinearity plot of *SlMDH* genes between *Arabidopsis*, and rice plant species. The chromosomes of the three species are represented in different colors: tomato, blue; *Arabidopsis*, green; and rice, dark yellow. All chromosomes are drawn to scale (in Mb). Gray lines in the background show collinear blocks within tomato and other plant genomes, while red lines indicate syntenic *MDH* gene pairs. The red lines between two chromosomal locations indicate a syntenic relationship between tomato (Sl-1 to 12) and *Arabidopsis* (1 to 5)/ Rice (1 to 12).

**Figure 2 ijms-23-10028-f002:**
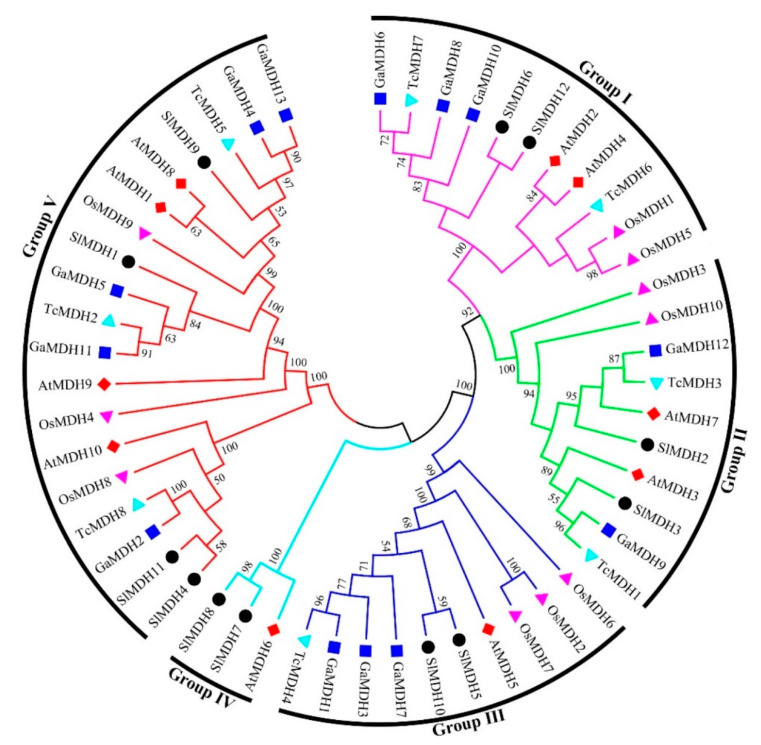
Phylogenetic relationships of MDHs from *Solanum lycopersicum*, *Arabidopsis thaliana*, *Oryza sativa*, *Gossypium arboretum and Theobroma cacao*. The un-rooted phylogenetic tree was generated with MEGA 6 using the neighbor-joining (NJ) method, and the bootstrap analysis was carried out with a total of 1000 replicates. The *MDHs* genes from *Solanum lycopersicum*, *Arabidopsis thaliana*, *Oryza sativa*, *Gossypium arboretum, and Theobroma cacao* were marked with a black circle, red rhombus, pink triangle, blue square, and cyan blue triangle, respectively.

**Figure 3 ijms-23-10028-f003:**
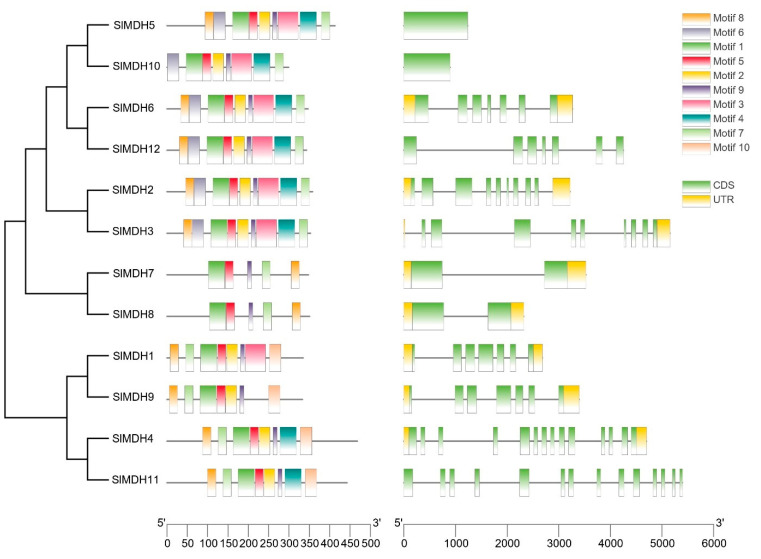
Motif identification (**left panel**) and gene structure (**right panel**) analysis of tomato MDHs. Each motif is shown in a different color. The intron/exon structure of tomato MDH genes. Exons and introns are shown by green boxes and grey lines, respectively.

**Figure 4 ijms-23-10028-f004:**
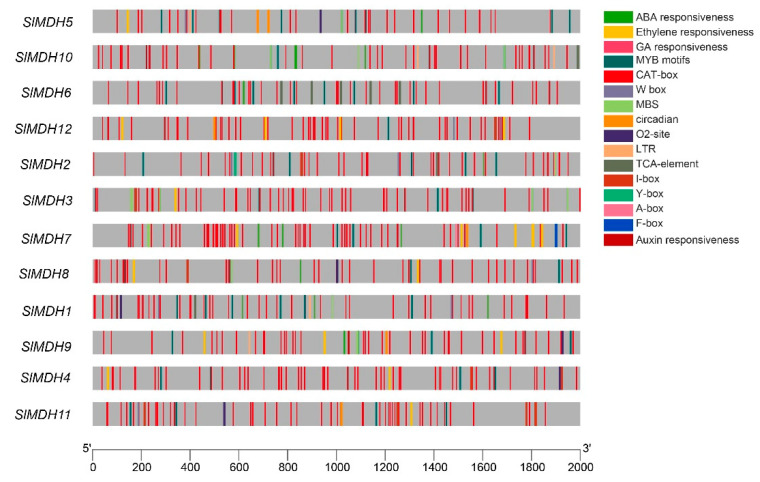
*Cis*-regulatory motifs found in the promoters of *SlMDHs*. *Cis*-elements are represented by various colored rectangles.

**Figure 5 ijms-23-10028-f005:**
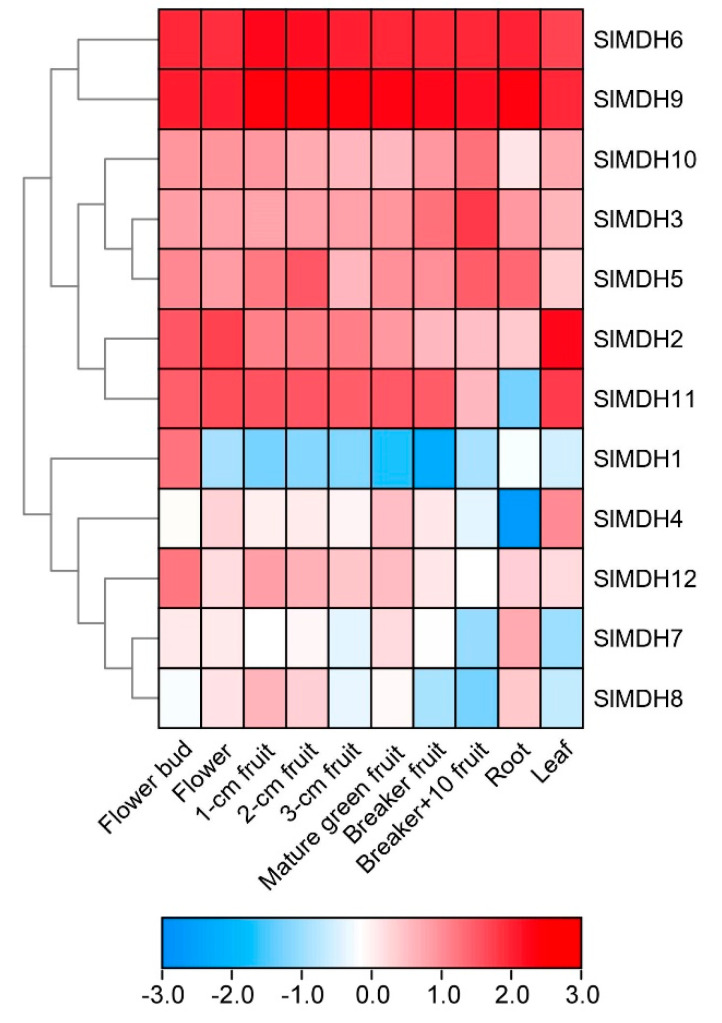
Expression analysis of tomato *MDH* genes in cultivated tomato cultivar *Heinz* 1706. Heatmap of RNA-seq data of *Heinz.* Flower bud, flower (fully opened flowers), 1-cm fruit, 2-cm fruit, 3-cm fruit, mature green fruit, breaker fruit, fruit at 10 days after breaker stage (Breaker+10 fruit), root, and leaf. The expression values were calculated by Log^2^ (FPKM) and presented according to the color code.

**Figure 6 ijms-23-10028-f006:**
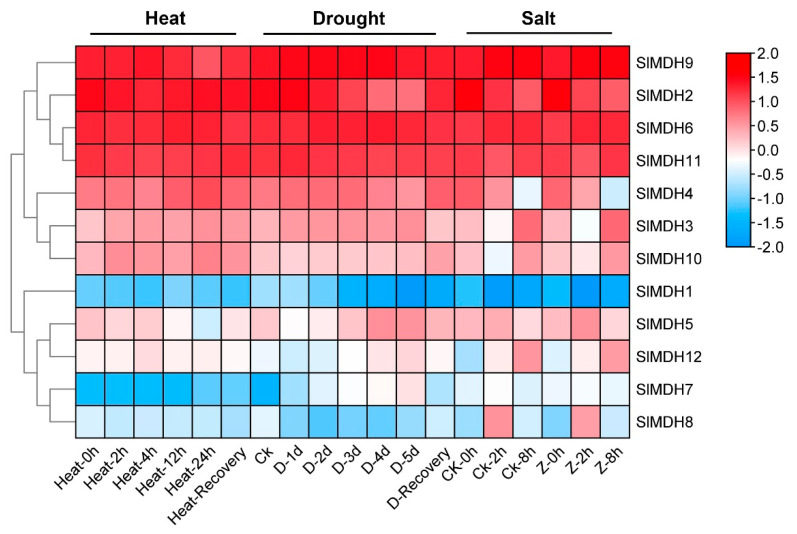
Gene expression analysis of tomato *MDH* genes under heat, drought, and salt stress using RNA-sequencing data. “D” and “Z” indicate drought and salt, respectively. The expression values were calculated by Log^2^ (FPKM) and presented according to the color code.

**Figure 7 ijms-23-10028-f007:**
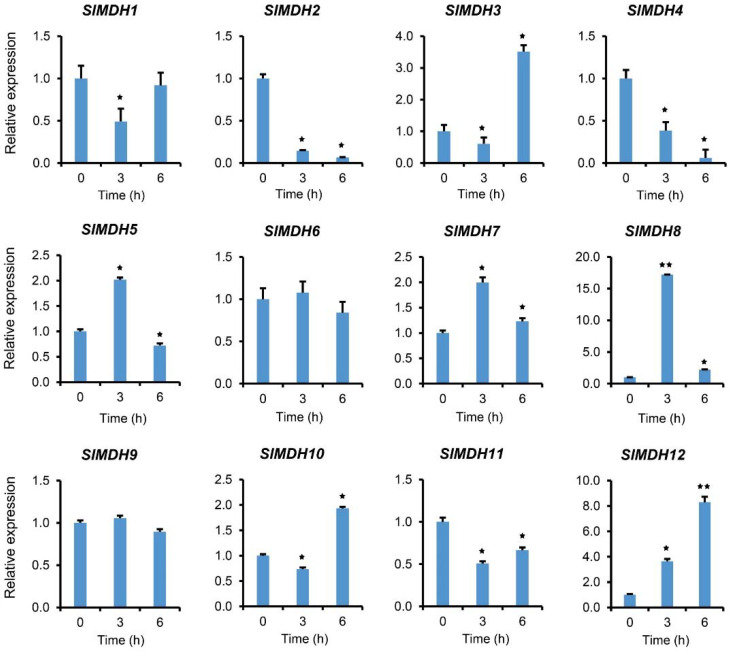
Gene expression analysis of tomato *MDH* genes in response to salt stress at different time points “0, 3, and 6 h” under 250 mM NaCl treatments. The relative expressions at different stress treatment times were compared with the control (0 h) and the control was set 1 to count fold change expression. *Actin* was used to normalize the data. Three biological replicates were used in the experiment. Error bars denote standard errors of the means of three independent technical replicates. The asterisks indicate significant differences, as determined by Student’s *t*-test (* *p*-value ≤ 0.05, **** and *p*-value ≤ 0.01).

**Figure 8 ijms-23-10028-f008:**
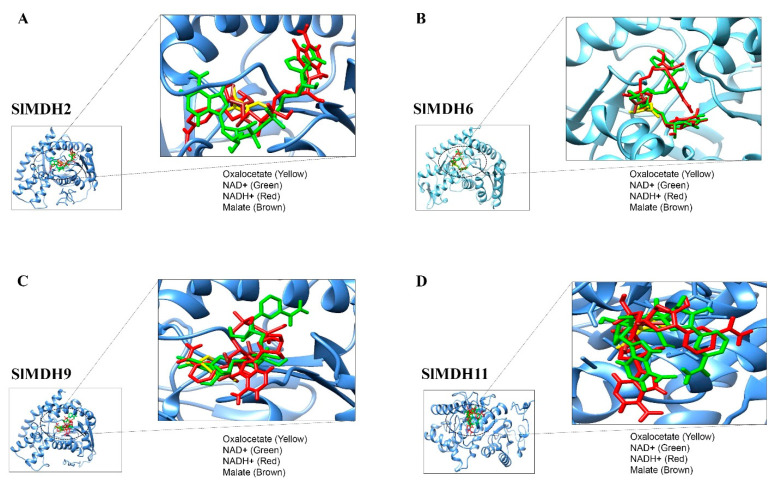
Molecular docking of Oxaloacetate (Yellow), NAD+ (Green), Malate (Brown) and NADH (Red) with SIMDH2 (**A**), SIMDH6 (**B**), SIMDH9 (**C**), SIMDH11 (**D**). The enzyme is shown in a blue cartoon while ligands are presented in different colors of sticks.

**Figure 9 ijms-23-10028-f009:**
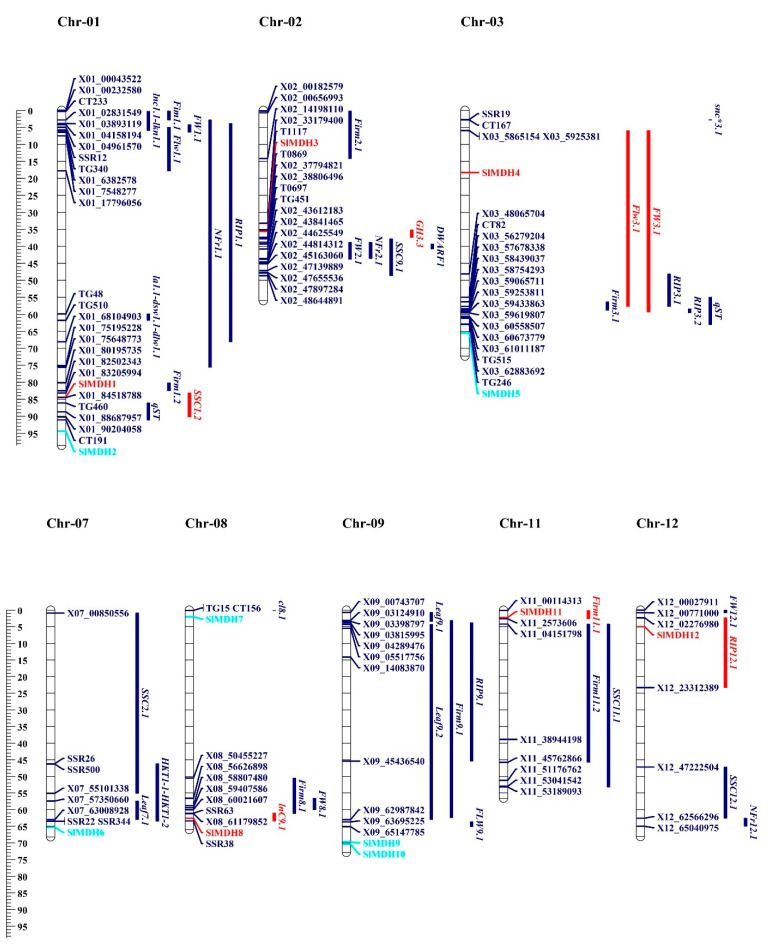
On the chromosomes of tomato, the *MDH* genes and the quantitative trait loci (QTLs) responsible for salt stress-related traits were found to be in close proximity to one another. This scale illustrates the relative physical locations of genes and QTL-linked markers in megabases (Mb). Genes located within the QTLs are illustrated with red color, while genes and QTLs not co-localized are represented with turquoise and blue colors.

**Table 1 ijms-23-10028-t001:** 12 *MDH* genes retrieved sequences from the genome of *S. lycopersicum*.

Gene ID	Gene Name	Genome Positioin	CDS	Size (aa)	MW (kDa)	pI	Subcellular Localization
Solyc01g090710.2.1.ITAG2.4	*SlMDH1*	ch01:84349916..84352603 forward	1005	334	35.88	6.46	Cytoplasmic
Solyc01g106480.2.1.ITAG2.4	*SlMDH2*	ch01:94360820..94364041 reverse	1074	357	37.63	8.1	Glyoxysomal
Solyc02g063490.2.1.ITAG2.4	*SlMDH3*	ch02:35565021..35570176 reverse	1059	352	36.85	8.13	Chloroplast
Solyc03g071590.2.1.ITAG2.4	*SlMDH4*	ch03:18254318..18259019 reverse	1404	467	51.66	7.89	Chloroplast
Solyc03g115990.1.1.ITAG2.4	*SlMDH5*	ch03:65541129..65542368 forward	1239	412	43.19	8.34	Chloroplast
Solyc07g062650.2.1.ITAG2.4	*SlMDH6*	ch07:65338276..65341542 forward	1041	346	36.08	8.73	Mitochondrial
Solyc08g007420.2.1.ITAG2.4	*SlMDH7*	ch08:1979674..1983199 forward	1044	347	37.64	5.36	Cytoplasmic
Solyc08g078850.2.1.ITAG2.4	*SlMDH8*	ch08:62549883..62552206 forward	1053	350	37.7	6.31	Chloroplast
Solyc09g090140.2.1.ITAG2.4	*SlMDH9*	ch09:69683418..69686815 reverse	999	332	35.38	5.91	Cytoplasmic
Solyc09g091070.1.1.ITAG2.4	*SlMDH10*	ch09:70406056..70406953 forward	897	298	31.81	5.35	Cytoplasmic
Solyc11g007990.1.1.ITAG2.4	*SlMDH11*	ch11:2198504..2203902 forward	1329	442	48.44	6.23	Chloroplast
Solyc12g014180.1.1.ITAG2.4	*SlMDH12*	ch12:5027816..5032069 reverse	1029	342	35.65	8.9	Mitochondrial

**Table 2 ijms-23-10028-t002:** Molecular docking results of the selected tomato proteins with different substrates (malate and oxaloacetate) and cofactors (NAD+ and NADH).

Complex	Binding Affinity (kcal/mol)	No. of Hydrogen Bond	Interacting Residues
**SIMDH11 Docking Results**			
SIMDH11 with Oxaloacetate	−3.7	A: Asp198–UNK: O A: Cys418–UNK: OH A: His421–UNK: O	Cys418, Asp198, Gly195
SIMDH11 with NAD+.	−3.9	A: Gln202–UNK: O A: Gly195–UNK: O	Leu197, Ala428, Gln202, Cys430
SIMDH11 with Malate.	−3.7	A: His421–UNK: O A: His421–UNK: OH A: Cys418–UNK: O A: Cys418–UNK: OH A: Asp198–UNK: O	Ala428, Cys418, Ala194, Gly195, Leu197
SIMDH11 with NADH.	−7.5	A: His421–UNK: O	Cyc430, Glu192, Leu197, Glu415, Arg417, Ile 199
**SIMDH6 Docking Results**			
SIMDH6 with Oxaloacetate	−4.6	A: Gly232–UNK:O A: Arg163–UNK: O A: Arg163–UNK: O A: Gln192–UNK: O A: Asp160–UNK: OH A: His188–UNK: OH	Ile236, Gln229, Gln230, Ser189, Asp160
SIMDH6 with NAD+.	−9.9	A: Asn187–UNK: O A: Arg99–UNK: O A: Asn132–UNK: OH A: Asn32–UNK: OH A: Gln192–UNK: O A: His188–UNK: NH A: Arg163–UNK: O A: Ser242–UNK: O A: Val130–UNK: NH A: Val130–UNK: NH	Glu323, Arg99, Gln192, Ser242, Gly232, Pro133, Ala131, Val102, Leu159
SIMDH6 with Malate.	−4.1	A: Ala233–UNK: O A: Gly232–UNK: O A: Arg163–UNK: O A: Gln192–UNK: O	Gln230, Gly232, Ser189, Asp160, Ser190
SIMDH11 with NADH.	−10	A: Ser190–UNK: O A: Gln192–UNK: O A: Arg163–UNK: O A: Val130–UNK: NH A: Leu156–UNK: NH	Arg93, Gln192, Ser243, Val130, Lew159, Pro133, Ile236, Ile235, Asn132
**SIMDH2 Docking Results**			
SIMDH2 with Oxaloacetate	−4.7	A: Asn138–UNK: O A: Asn163–UNK: O	Ala121, Gly122, Ile119, Asn163, Pro124
SIMDH2 with NAD+.	−8.3	A: Val79–UNK: O A: Met272–UNK: O A: Ile123–UNK: NH A: Ile123–UNK: NH	Tyr77, Gly52, Thr269, Ile161, Ile123, Arg125, Gly122, Ala121
SIMDH2 with Malate.	−4.4	A: Asn138–UNK: O	Gly122, Asn163, Ile57, Pro120, Thr269, Asn138
SIMDH2 with NADH.	−8.3	A: Asn163–UNK: O A: Ile57–UNK: O A: Gly56–UNK: O A: Asp78–UNK: O	Leu193, Ala268, Asn138, Pro120, Thr269, Tyr77, Asp78, Asn163
**SIMDH9 Docking Results**			
SIMDH9 with Oxaloacetate	−4.7	A: Ile45–UNK: O A: Ile45–UNK: O A: Gly46–UNK: O A: Gly110–UNK: O A: Pro108–UNK: OH	Gly44. Pro112, Ser255, Pro108, Gly110, Gly46, Ile45
SIMDH9 with NAD+.	−8.3	A: Gly110–UNK: OH A: Ile45–UNK: O A: Ile149–UNK: NH	Tyr65, Gly110, Asp66, Gly43, Ile45, Pro112, Val111, Ile149, Ile129
SIMDH9 with Malate	−4.2	A: Ile145–UNK: O A: Gly46–UNK: O A: Ile149–UNK: NH A: Ile149–UNK: NH	Gly46, Gly43, Ala109, Gly110, Ala109
SIMDH9 with NADH	−8.6	A: Gly40–UNK: OH A: Gly43–UNK: OH A: Asp66–UNK: OH A: Ile67–UNK: O	Gly40, Asp66, Val111, Ala109, Pro108, Pro112, Ala42

Abbreviations: Asparagine—(Asn), Aspartic acid—(Asp), Cysteine—(Cys), Glutamic acid (Glu), Glutamine—(Gln), Glycine—(Gly), Histidine—(His), Isoleucine—(Ile), Leucine—(Leu), Lysine—(Lys), Methionine—(Met), Phenylalanine—(Phe), Proline—(Pro), Serine—(Ser), Threonine—(Thr), Tryptophan—(Trp), Tyrosine—(Tyr), Valine—(Val).

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
