# Peer review of "Identification and Characterization of Malate Dehydrogenases in Tomato (Solanum lycopersicum L.)"

_ijms, 2022, doi:10.3390/ijms231710028_

Round 1
Reviewer 1 Report
The manuscript entitled “Genomic Identification, Molecular Characterization and QTL Mapping of Malate Dehydrogenase Genes confers SlMDH11 is Potential Candidate for Salt Tolerance in Tomato” is an interesting work. The Authors identified and characterized tomato malate dehydrogenase (MDH) genes and encoded by them proteins known to be responsible for the reversible conversion of malate to oxaloacetate. The Authors presented results, which were obtained applying in silico tools (broad range) and by carrying out qRT-PCRs experiments.
Although the work is quite interesting, it needs further improvement. Comment and suggestions are written below.
General comments:
1. Please, make sure to write gene names always in italics, including in figures (whenever possible; e.g. Figure 4) or when you write MDH gene family, MDH group, etc.
2. While applying the style with starting a word with a big letter, please, make sure where to use them and when not.
3. I strongly suggest to write the part Materials and Methods with more details.
4. Improve English.
Title
I would be very careful with using the term “OTL mapping” in the title as you did not perform a typical QTL mapping study.
I think “a Potential Candidate”.
Affiliations
Please, try to edit affiliation part so that it is uniform.
Please mark the meaning of “+”.
Abstract
Line 25
After Solanum lycopersicum write L. or Mill.
Line 27
“The tomato MDH genes were split into five groups based on phylogenetic analysis, with genes that clustered together sharing the same subcellular position and having similar sequence lengths, gene structures, and conserved motifs.”
That is not fully correct as e.g.: genes in the group II localize to chloroplasts and glyoxysomes, in the group III to chloroplasts and cytosol, and so on.
Line 33
“at various stages” – here and in other places in the manuscript
Line 35
I would not come to this conclusion that “especially salt stress”. I suggest to remove it.
Line 36
There should not be “,” after “to abiotic stresses”.
1. Introduction:
Line 50
I would be very careful here. I suggest to remove “all”. In the publication that you cite and e.g. in [5] it states most of the bacteria species, not all. In general, I suggest to place here a more updated reference.
Line 56
I believe to write “The active site of MDH proteins is located in a cleft between the two domains, and each subunit has two distinct domains: a NAD/NADP-binding domain and a substrate-binding domain [4, 5].” is not clear. Thus, I suggest to first write about domains in each subdomain and then about the active site between these two domains.
Line 60-64
Citations
Line 67
Citation for each species
Line 68
I believe you meant 13, 13 and 25.
Line 73
MDH2 not mDH2
Line 74-76
Since you write about the engagement of MDH proteins in plant responses to high salt concentration later, at the end of this page, the following information “The overexpression of NADP-74 MDH can enhance salt tolerance in Arabidopsis thaliana [12]. Likewise, apple cytosolic MDH can increase the cold and salt tolerance [13, 14].” could be shortened here. You could just state that MDH proteins were shown to be involved in plant responses to different abiotic stresses, e.g. cold, salinity, and leave the citations that you have at the moment. The more exact information could be added to the part about salt stress at the end of this page.
Line 76
“As a result” does not fit here. I suggest to write e.g. “In summary”, and not “over the evolutionary process of different species” but e. g. “over the course of evolution”.
Line 80
After Solanum lycopersicum write L. or Mill.
Tomato is not a model plant for the production of fleshy fruits but rather for studying fleshy fruits.
Line 82
I suggest to cite the size of the tomato gnome with some more updated source.
Line 83
I believe you wanted to write that tomato plants have a relatively short life cycle, not the genome of the species. Please, try to correct it.
Line 86
The citation [17] is the same as [15]. Moreover, if you want to write about tomato genome sequencing successes, I suggest to use the most updated sources (not from 2012).
Line 86-87
“Despite the fact that tomato production has been steadily increasing in recent years,” – please, add citation.
Line 92
Once you start to write about MDH, I suggest to start it with a new sentence.
Line 94-95
Just before writing the following sentence: “For instance, rice plastidial NAD-dependent MDH1 reduces the response to salt stress by modifying the vitamin B6 concentration of rice tissues [18].” you mention about the identification of possible gene that may confer resistance to abiotic stresses. The way you write is contradictory. I suggest to better write about genes that regulate plant responses to high salt concentration, not confer resistance. Then you can write that MDH proteins were found as both positive and negative regulators of plant responses to salt stress. Eventually, you could enumerate the examples, including the ones from lines 75-76.
Results
Line 119-121
“The candidate MDH genes were identified through a systematic BLAST search against the tomato genome database with query sequences of G. arboretum and Arabidopsis.”
It does not correspond with what is written in Materials and Methods (lines 331-334):
“Solanum lycopersicon MDH family sequences were retrieved from Phytozome v12.1 [33]. Arabidopsis thaliana, Gossypium hirsutum and Oryza sativa sequences obtained from Phytozome v12.1 database, were used as query sequences.
Please, write in details the steps taken. Write also exactly what kind of sequences did you use as queries. I believe not all Arabidopsis thaliana, Gossypium hirsutum and Oryza sativa MDH sequences? Please, specify what type of sequences were used – amino acid, nucleotide (best in Materials and Methods section).
Line 127
Instead of “investigations” I suggest to write “parameters”.
Line 131-134
“The number of amino acids contained in SlMDH proteins 131 varied in length, going anywhere from 298 to 467. With a mean value of 38.99 132 KDa, the molecular weights of MDH proteins ranged anywhere from 31.81 133 to 51.66 KDa.” – it is a repetition of what is written in the lines before in your manuscript.
Line 137
Please, write in Materials and Methods the program that you used to assess protein localization.
Line 1– 45
This part needs quite a lot of improvement. It has to be better explained and written.
Line 13-14
After the following sentence: “In addition, during the process of tomato domestication, both whole-genome duplication (WGD) and segmental duplication occurred, both of which had an essential part in the expansion of the apple gene family.” please, write citation.
Could you also be more specific with the apple gene family (which genes?), and why do you write about apple. That has to be explained.
Line 16
I believe “the growth” in “the growth of SlMDH genes” is not the best word to use.
Line 19
“The Ka/Ks ratio is an important measure for determining the number of replication events as well as the selection forces [21].” – do you mean here duplication events?
Line 20-22
“We calculated evolutionary rates and selective pressure by using the Ka (nonsynonymous), Ks (synonymous),” – try to explain it better, adding some information about substitution per site.
Line 24-26
“Our goal was to determine whether or not a SlMDH gene is under different selection pressures and has different evolutionary rates than its duplicated genes [21].” - it is a repetition of what is written in the lines before in your manuscript.
Line 26-28
“As a result, we studied the selective pressures that are exerted on duplicated MDH genes; the values of each duplicated gene pair are presented.” – it is not clear where are the values presented.
Line 30
What do you mean by writing more pure forms?
Line 31-32
Sentence not grammatically correct.
Line 35-36
“the evolutionary mechanism of SlMDH members.” – it is not correct. SlMDH members do not have an evolutionary mechanism.
Line 36-43
All information about particular genes should be placed in the figure 1A and 1B. Otherwise, it is hard to analyze such data.
Line 53 (Figure 1 description)
“show” instead of “showed”
Line 55-80
I suggest to create e. g. a supplementary file with information on accession numbers for all sequences used to create the phylogenetic tree. It also has to be clearly stated in the Material and Methods section how did you obtain these sequences.
Line 66-69
“and it was found that group V had a greater number of members than the other three groups combined. In addition to this, it was found that the SlMDHs have a greater evolutionary link with the AtMDHs and the GaMDHs within each group.”
I suggest to remove that part. First element is not important, and the last sentence is not correct. I suggest just to explain into which group fall particular tomato MDH protein.
Line 72
The figure 2 is not split into subfigures. Thus, I recommend to remove the letter “A” from the figure description part.
Line 77
You could write not just about colors but also about figures that were chosen for particular species.
Line 84-88
Since here it is not easy to follow, which SlMDH proteins fall into particular group, I suggest to write names of SlMDH in brackets when writing about groups. Moreover, earlier to describe these groups you used Roman letters, and here Arabic. Please, try to make it more uniform.
Line 85
“while motif 3 and motif 7 are absent in Group 5” – that is not fully true as SlMDH1 has motif 3.
Line 87
„motif 10 is only present in Group 4 and Group 5,” – again, not correct. This motif is present only in group 5.
Line 98
Figure 3, not 5.
Line 99
“in SlMDH genes ranged from 0 to 13, with SlMDH11 having the most introns,” – I guess both 11 and 4?
Line 104-116
I suggest to write more information about the particular cis-regulatory elements found in the promoter regions.
Line 105
Instead of writing “in the upstream 2 kb promoter regions of SlMDH genes” I suggest to write upstream of the transcription start site.
Line 108
Are you sure about CAT boxes? Aren’t they CAAT boxes? (here and in the figure).
There are no TATA boxes presented in the figure.
Line 118
All promoters have some regulatory sequences. The sentence has to be rewritten.
Line 121
“tissues” not “tissue”
Line 121-134
First, I suggest to place the SF2 in the main text and to describe it in details, especially particular stages as it is not fully clear, e,g, what is the age of flowers (days after pollination), what means Breaker +10, etc.
Please, correct also the description in the text as not all the information that you write about the results is correct in relation to what is presented in the SF2.
Line 136-148
Either in the section Materials and Methods or here, describe in more details the treatments.
Please, correct the description for the results presented in the figure 5 as not all the information that you write about these results is true. There is quite substantial number of mistakes. To avoid such mistakes, I suggest to not write about the results in a too general way.
List the 7 genes that you write about,
Please, refer to the Recovery phase in the figure 5. What does it mean? Try to describe it more and explain these results as well.
Line 153-155
Please, describe all abbreviations used in the figure. Moreover, I strongly suggest to make the scale more uniform, e.g. from -1 to 3 in all figures, and for “-“ results use the blue color, for “0” white, and for the “+” red. At this point it is quite confusing. I think some confusion in describing these results may results from this not uniform color code.
Not “from RNA-sequencing” but e.g. “based on RNA-sequencing data” or “using RNA-sequencing data”
Line 159
Not “by exposing tomato plants to salt stress” but “on plants exposed to salt stress”
Line 174
Statistical analysis of difference significance is needed for the gene expression studies.
Line 183-184
There is no information on residues in Figure 7. It would be actually good to add such information.
Line 195
Correct the result for oxaloacetate
Line 194-195
“However, the NADH molecule showed the strongest binding with all the SlMDHs proteins” – not correct, for SlMDH2 – the same for NAD and NADH
Line 204
“2, 4 and 3 hydrogen bonds” not “2, 4 and 2 hydrogen bonds”
Line 219
What about NADP? How do you know which MDH has NAD+ or NADP+ cofactor?
It is known from kinetic studies that the malate to oxaloacetate reaction is an ordered reaction with NAD/NADH binding first, followed by the dicarboxylic acid substrate (Silverstein et al. 1969). It would mean that the docking for malate has to be done for the MDH molecule with the cofactor. Is it possible to do it?
Please, describe the meaning for all abbreviations used in the table.
Line 222-225
What are these ligands exactly with particular SlMDH? Please, add that information.
I suggest to write just “blue” as there are different blue colors for proteins in figure 7.
Line 228-237
Please, cite all the QTL studies that were used by you to mark QTLs in Figure 8. I also suggest to at least write their names while describing them in the text.
Discussion
Line 254
“MDHs are very…” not “MDH is a very”
Line 254-263
There are some unnecessary repetitions, and quite some grammatic mistakes.
Citation [24] is not about MDHs.
Line 268
These genes are not localized in chloroplasts but their products.
Four groups not two.
Line 270
I am already confused whether you analyzed the motifs using gene sequences or amino acid ones. Here you write genes, in lines 80-82 protein sequences, and then in Material and Methods (4.2) – genes. That has to be clarified.
Line 274
You did not show or write such results. I suggest to transfer this information to the results part.
Line 296
Please, write names of the duplicated genes.
Line 302-303
There were no 3 groups for the RNA-seq data.
Line 304-307
You have to be mor specific. That does not apply to all the genes studied.
Line 307-308
There is no previous information (and in Materials and Methods) that the expression was analyzed in roots. All this has to be more explained.
Line 310
“especially the SlMDH2, and SlMDH11 gene at 3hr” – it is not correct – 2 and 12.
Line 315-317
This information is not correct.
Line 324-325
“Our docking results showed that binding affinity of SlMDH11 is similar between oxaloacetate and malate,” – it was similar for other analyzed SlMDHs as well
In my opinion there are no reasons to somehow say that from all the identified and characterized MDHs in tomato, the 11th one is somehow more involved in salt stress responses. Thus, the title has to be also adjusted accordingly.
Material and Methods
Line 334
BLAST is a tool not a verb.
Line 335
“With the help of ProtParam Tool,” – please, add citation.
Line 343
“motifs were selected as 10 and the other parameters used were default” – instead, I suggest to write “motifs were selected as 10 and the other parameters were used as default”
Line 346
Not “4.3 Phylogenetic analysis and sequence alignment” but “4.3 Sequence alignment and phylogenetic analysis”
Line 350
Which MEGA was used – 6 or 7? (somewhere before you wrote 6).
Line 350-351
Try to combine it with the first sentence in this part of the text.
Line 355
The name of the program is PlantCARE. Please, cite it.
Line 358-365
This part – “Synteny and collinearity analysis” – should be placed before “4.3 Phylogenetic analysis and sequence alignment”. Part 4.2 should go after the Phylogenetic one.
Line 367-377
Cite the databases. I believe more information on treatments has to be written.
Line 381
The tomato name has to be editd.
Line 382
Seeds were sown and plant were grown at
Line 384
How old were these seedlings?
Line 385
Please, write the composition of the solution
Line 386-388
What was the volume of the original solution and then how much of the 250 mM solution was added?
Line 391
You do not show results for these timepoints.
Line 396
To the list in the ST2, please, add primers for actine.
Line 397
Please, add citation for the method used to calculate relative expression.
Line 404
Citations for the sources
References
Please, remember about writing Latin names of species in italics.
15. I guess it is better to write as authors “Sato S,…. and the other authors (till the number of authors allowed by IJMS). The “x” is I guess not needed.
34. I believe there are some mistakes in the citation.
46. The paper that you cite is just a review. I suggest then to rather cite the program with the website of use, according to the standards of IJMS. Otherwise, I believe the review should be cited including following information: “J. Chem. Inf. Model. 2010, 50, 11, 2053. “ (in a way adjusted to the requirements of IJMS).

Author Response
Reviewer
The manuscript entitled “Genomic Identification, Molecular Characterization and QTL Mapping of Malate Dehydrogenase Genes confers SlMDH11 is Potential Candidate for Salt Tolerance in Tomato” is an interesting work. The Authors identified and characterized tomato malate dehydrogenase (MDH) genes and encoded by them proteins known to be responsible for the reversible conversion of malate to oxaloacetate. The Authors presented results, which were obtained applying in silico tools (broad range) and by carrying out qRT-PCRs experiments.
Although the work is quite interesting, it needs further improvement. Comment and suggestions are written below.
We thank the reviewer for suggestions/comments that helped us a lot to improve the quality of the manuscript.
General comments:
- Please, make sure to write gene names always in italics, including in figures (whenever possible; e.g. Figure 4) or when you write MDHgene family, MDH group, etc.
Thank you. We modified the text as suggested.
- While applying the style with starting a word with a big letter, please, make sure where to use them and when not.
Thank you. We modified the text as suggested.
- I strongly suggest to write the part Materials and Methods with more details.
Thank you. We elaborated our material and methods.
- Improve English.
Thank you. We carefully read the manuscript.
Title
I would be very careful with using the term “OTL mapping” in the title as you did not perform a typical QTL mapping study.
I think “a Potential Candidate”.
Thank you. We changed the title.
Affiliations
Please, try to edit affiliation part so that it is uniform.
Please mark the meaning of “+”.
Thank you. We edited the affiliation so that they are uniform.
Abstract
Line 25
After Solanum lycopersicum write L. or Mill.
Thank you. We modified the text as suggested.
Line 27
“The tomato MDH genes were split into five groups based on phylogenetic analysis, with genes that clustered together sharing the same subcellular position and having similar sequence lengths, gene structures, and conserved motifs.”
That is not fully correct as e.g.: genes in the group II localize to chloroplasts and glyoxysomes, in the group III to chloroplasts and cytosol, and so on.
Thank you. We modified the text.
Line 33
“at various stages” – here and in other places in the manuscript
Thank you. We modified the text.
Line 35
I would not come to this conclusion that “especially salt stress”. I suggest to remove it.
Thank you. We modified the text as suggested.
Line 36
There should not be “,” after “to abiotic stresses”.
Thank you. We modified the text as suggested.
- Introduction:
Line 50
I would be very careful here. I suggest to remove “all”. In the publication that you cite and e.g. in [5] it states most of the bacteria species, not all. In general, I suggest to place here a more updated reference.
Line 56
I believe to write “The active site of MDH proteins is located in a cleft between the two domains, and each subunit has two distinct domains: a NAD/NADP-binding domain and a substrate-binding domain [4, 5].” is not clear. Thus, I suggest to first write about domains in each subdomain and then about the active site between these two domains.
Thank you. We modified the text as suggested.
Line 60-64
Citations
Thank you. We added the citations.
Line 67
Citation for each species
Thank you. We added the citations.
Line 68
I believe you meant 13, 13 and 25.
Thank you. Yes, it should be 13, 13, and 25 and modified the text.
Line 73
MDH2 not mDH2
Thank you. It was mitochondrial MDH2 and corrected the text.
Line 74-76
Since you write about the engagement of MDH proteins in plant responses to high salt concentration later, at the end of this page, the following information “The overexpression of NADP-74 MDH can enhance salt tolerance in Arabidopsis thaliana [12]. Likewise, apple cytosolic MDH can increase the cold and salt tolerance [13, 14].” could be shortened here. You could just state that MDH proteins were shown to be involved in plant responses to different abiotic stresses, e.g. cold, salinity, and leave the citations that you have at the moment. The more exact information could be added to the part about salt stress at the end of this page.
Thank you. We modified the text as suggested.
Line 76
“As a result” does not fit here. I suggest to write e.g. “In summary”, and not “over the evolutionary process of different species” but e. g. “over the course of evolution”.
Thank you. We modified the text as suggested.
Line 80
After Solanum lycopersicum write L. or Mill.
Tomato is not a model plant for the production of fleshy fruits but rather for studying fleshy fruits.
Thank you. We modified the text as suggested.
Line 82
I suggest to cite the size of the tomato gnome with some more updated source.
Thank you. We modified the text.
Line 83
I believe you wanted to write that tomato plants have a relatively short life cycle, not the genome of the species. Please, try to correct it.
Thank you. We modified the text.
Line 86
The citation [17] is the same as [15]. Moreover, if you want to write about tomato genome sequencing successes, I suggest to use the most updated sources (not from 2012).
Thank you. We are not focusing on tomato genome sequencing history, therefore, we modified the text.
Line 86-87
“Despite the fact that tomato production has been steadily increasing in recent years,” – please, add citation.
Thank you. We added the citation.
Line 92
Once you start to write about MDH, I suggest to start it with a new sentence.
Thank you. We modified the text as suggested.
Line 94-95
Just before writing the following sentence: “For instance, rice plastidial NAD-dependent MDH1 reduces the response to salt stress by modifying the vitamin B6 concentration of rice tissues [18].” you mention about the identification of possible gene that may confer resistance to abiotic stresses. The way you write is contradictory. I suggest to better write about genes that regulate plant responses to high salt concentration, not confer resistance. Then you can write that MDH proteins were found as both positive and negative regulators of plant responses to salt stress. Eventually, you could enumerate the examples, including the ones from lines 75-76.
Thank you. We modified the text as suggested.
Results
Line 119-121
“The candidate MDH genes were identified through a systematic BLAST search against the tomato genome database with query sequences of G. arboretum and Arabidopsis.”
It does not correspond with what is written in Materials and Methods (lines 331-334):
“Solanum lycopersicon MDH family sequences were retrieved from Phytozome v12.1 [33]. Arabidopsis thaliana, Gossypium hirsutum and Oryza sativa sequences obtained from Phytozome v12.1 database, were used as query sequences.
Please, write in details the steps taken. Write also exactly what kind of sequences did you use as queries. I believe not all Arabidopsis thaliana, Gossypium hirsutum and Oryza sativa MDH sequences? Please, specify what type of sequences were used – amino acid, nucleotide (best in Materials and Methods section).
Thank you. We modified the text as suggested.
Line 127
Instead of “investigations” I suggest to write “parameters”.
Thank you. We modified the text as suggested.
Line 131-134
“The number of amino acids contained in SlMDH proteins 131 varied in length, going anywhere from 298 to 467. With a mean value of 38.99 132 KDa, the molecular weights of MDH proteins ranged anywhere from 31.81 133 to 51.66 KDa.” – it is a repetition of what is written in the lines before in your manuscript.
Thank you. We modified the text.
Line 137
Please, write in Materials and Methods the program that you used to assess protein localization.
Thank you. We added the information in material and methods.
Line 1– 45
This part needs quite a lot of improvement. It has to be better explained and written.
Thank you. We modified the text as suggested.
Line 13-14
After the following sentence: “In addition, during the process of tomato domestication, both whole-genome duplication (WGD) and segmental duplication occurred, both of which had an essential part in the expansion of the apple gene family.” please, write citation.
Could you also be more specific with the apple gene family (which genes?), and why do you write about apple. That has to be explained.
Thank you. We modified the text.
Line 16
I believe “the growth” in “the growth of SlMDH genes” is not the best word to use.
Thank you. We modified the text.
Line 19
“The Ka/Ks ratio is an important measure for determining the number of replication events as well as the selection forces [21].” – do you mean here duplication events?
Thank you. Yes, it is duplication of events and corrected the text.
Line 20-22
“We calculated evolutionary rates and selective pressure by using the Ka (nonsynonymous), Ks (synonymous),” – try to explain it better, adding some information about substitution per site.
Thank you. We modified the text as suggested.
Line 24-26
“Our goal was to determine whether or not a SlMDH gene is under different selection pressures and has different evolutionary rates than its duplicated genes [21].” - it is a repetition of what is written in the lines before in your manuscript.
Thank you. We removed the repetitions in the text as suggested.
Line 26-28
“As a result, we studied the selective pressures that are exerted on duplicated MDH genes; the values of each duplicated gene pair are presented.” – it is not clear where are the values presented.
Thank you. We mentioned Table S2 in the text.
Line 30
What do you mean by writing more pure forms?
Thank you. We modified the text.
Line 31-32
Sentence not grammatically correct.
Thank you. We modified the text.
Line 35-36
“the evolutionary mechanism of SlMDH members.” – it is not correct. SlMDH members do not have an evolutionary mechanism.
Thank you. We modified the text.
Line 36-43
All information about particular genes should be placed in the figure 1A and 1B. Otherwise, it is hard to analyze such data.
Thank you. We have written the information in the text very clear, but it’s hard and create overlapping of genes. Therefore, we used the figures as such obtained by the software.
Line 53 (Figure 1 description)
“show” instead of “showed”
Thank you. We corrected the text.
Line 55-80
I suggest to create e. g. a supplementary file with information on accession numbers for all sequences used to create the phylogenetic tree. It also has to be clearly stated in the Material and Methods section how did you obtain these sequences.
Thank you. We added the supplementary file.
Line 66-69
“and it was found that group V had a greater number of members than the other three groups combined. In addition to this, it was found that the SlMDHs have a greater evolutionary link with the AtMDHs and the GaMDHs within each group.”
I suggest to remove that part. First element is not important, and the last sentence is not correct. I suggest just to explain into which group fall particular tomato MDH protein.
Thank you. We modified the text as suggested.
Line 72
The figure 2 is not split into subfigures. Thus, I recommend to remove the letter “A” from the figure description part.
Thank you. We corrected the text.
Line 77
You could write not just about colors but also about figures that were chosen for particular species.
Thank you. We modified the text as suggested.
Line 84-88
Since here it is not easy to follow, which SlMDH proteins fall into particular group, I suggest to write names of SlMDH in brackets when writing about groups. Moreover, earlier to describe these groups you used Roman letters, and here Arabic. Please, try to make it more uniform.
Thank you. We corrected the text as suggested.
Line 85
“while motif 3 and motif 7 are absent in Group 5” – that is not fully true as SlMDH1 has motif 3.
Thank you. We corrected the text.
Line 87
„motif 10 is only present in Group 4 and Group 5,” – again, not correct. This motif is present only in group 5.
Thank you. We corrected the text.
Line 98
Figure 3, not 5.
Thank you. We corrected the text.
Line 99
“in SlMDH genes ranged from 0 to 13, with SlMDH11 having the most introns,” – I guess both 11 and 4?
Thank you. We corrected the text.
Line 104-116
I suggest to write more information about the particular cis-regulatory elements found in the promoter regions.
Thank you. We modified the text.
Line 105
Instead of writing “in the upstream 2 kb promoter regions of SlMDH genes” I suggest to write upstream of the transcription start site.
Thank you. We modified the text.
Line 108
Are you sure about CAT boxes? Aren’t they CAAT boxes? (here and in the figure).
There are no TATA boxes presented in the figure.
Thank you. We modified the text, as CAAT and TATA box are commonly present in each promoter. Therefore, we did not mention them in the figure. However, we found the CAT-box and mentioned in the figure.
Line 118
All promoters have some regulatory sequences. The sentence has to be rewritten.
Thank you. We modified the text.
Line 121
“tissues” not “tissue”
Thank you. We corrected the text.
Line 121-134
First, I suggest to place the SF2 in the main text and to describe it in details, especially particular stages as it is not fully clear, e,g, what is the age of flowers (days after pollination), what means Breaker +10, etc.
Thank you. We brought the SF2 into the main text (Now Fig 5) and corrected the text. The detail of different stages are given in the figure legends.
Please, correct also the description in the text as not all the information that you write about the results is correct in relation to what is presented in the SF2.
Thank you. We corrected the text.
Line 136-148
Either in the section Materials and Methods or here, describe in more details the treatments.
Please, correct the description for the results presented in the figure 5 as not all the information that you write about these results is true. There is quite substantial number of mistakes. To avoid such mistakes, I suggest to not write about the results in a too general way.
Thank you. We modified the text.
List the 7 genes that you write about,
Thank you, We modified the text.
Please, refer to the Recovery phase in the figure 5. What does it mean? Try to describe it more and explain these results as well.
Thank you, We modified the text and added the information on the recovery phase.
Line 153-155
Please, describe all abbreviations used in the figure. Moreover, I strongly suggest to make the scale more uniform, e.g. from -1 to 3 in all figures, and for “-“ results use the blue color, for “0” white, and for the “+” red. At this point it is quite confusing. I think some confusion in describing these results may results from this not uniform color code.
Thank you, We modified the figure according to your suggestions.
Not “from RNA-sequencing” but e.g. “based on RNA-sequencing data” or “using RNA-sequencing data”
Thank you, We modified the text.
Line 159
Not “by exposing tomato plants to salt stress” but “on plants exposed to salt stress”
Thank you, We modified the text.
Line 174
Statistical analysis of difference significance is needed for the gene expression studies.
Thank you, We applied the T-test.
Line 183-184
There is no information on residues in Figure 7. It would be actually good to add such information.
Thank you. The interacting residues and binding information are already given in table 2. In addition, it is a lot easier to compare the four proteins in table 2.
Line 195
Correct the result for oxaloacetate
Thank you. we corrected the text.
Line 194-195
“However, the NADH molecule showed the strongest binding with all the SlMDHs proteins” – not correct, for SlMDH2 – the same for NAD and NADH
Thank you. we corrected the text.
Line 204
“2, 4 and 3 hydrogen bonds” not “2, 4 and 2 hydrogen bonds”
Thank you. we corrected the text.
Line 219
What about NADP? How do you know which MDH has NAD+ or NADP+ cofactor?
According to the malate valve hypothesis, NADPH is generated in the chloroplast through the electron transport chain during the day but does not readily diffuse out of the chloroplast. NADP-MDH reduces oxaloacetate to malate by oxidizing NADPH to NADP, and the malate is shuttled to the cytosol in exchange for oxaloacetate via the dicarboxylate transporter. In the cytosol, malate is oxidized back to oxaloacetate by the cytosolic MDHs, regenerating a reducing equivalent in the form of NADH in that compartment. So when the MDH is located in chloroplast it has NADP+ cofactor, while cytosolic MDH has NAD+ cofactor.
It is known from kinetic studies that the malate to oxaloacetate reaction is an ordered reaction with NAD/NADH binding first, followed by the dicarboxylic acid substrate (Silverstein et al. 1969). It would mean that the docking for malate has to be done for the MDH molecule with the cofactor. Is it possible to do it?
This is possible but it will need a lot of time as this is a very detailed study and requires a lot of computational resources. However, this is not the center of the current study, but we thank the reviewer for a nice suggestion.
Please, describe the meaning for all abbreviations used in the table.
Thank you. we modified the text by adding the abbreviations for each.
Line 222-225
What are these ligands exactly with particular SlMDH? Please, add that information.
I suggest to write just “blue” as there are different blue colors for proteins in figure 7.
Thank you. we modified the text.
Line 228-237
Please, cite all the QTL studies that were used by you to mark QTLs in Figure 8. I also suggest to at least write their names while describing them in the text.
Thank you. we cited the references and modified the text.
Discussion
Line 254
“MDHs are very…” not “MDH is a very”
Thank you. we modified the text.
Line 254-263
There are some unnecessary repetitions, and quite some grammatic mistakes.
Thank you. we modified the text.
Citation [24] is not about MDHs.
Thank you. we corrected this mistake.
Line 268
These genes are not localized in chloroplasts but their products.
Four groups not two.
Thank you. we removed the text as it was not fitting properly in the context of the paragraph here.
Line 270
I am already confused whether you analyzed the motifs using gene sequences or amino acid ones. Here you write genes, in lines 80-82 protein sequences, and then in Material and Methods (4.2) – genes. That has to be clarified.
We apologize for the inconvenience. We used the amino acid sequence for motifs, and we mentioned it in the text as well.
Line 274
You did not show or write such results. I suggest to transfer this information to the results part.
Thank you. we transfer this information in the text.
Line 296
Please, write names of the duplicated genes.
Thank you. we modified the text.
Line 302-303
There were no 3 groups for the RNA-seq data.
Thank you. we modified the text.
Line 304-307
You have to be mor specific. That does not apply to all the genes studied.
We apologize for the inconvenience. We removed the sentences it was misleading and did not properly present the facts.
Line 307-308
There is no previous information (and in Materials and Methods) that the expression was analyzed in roots. All this has to be more explained.
Thank you. we modified the text.
Line 310
“especially the SlMDH2, and SlMDH11 gene at 3hr” – it is not correct – 2 and 12.
Thank you. we corrected the text.
Line 315-317
This information is not correct.
Thank you. we corrected the text.
Line 324-325
“Our docking results showed that binding affinity of SlMDH11 is similar between oxaloacetate and malate,” – it was similar for other analyzed SlMDHs as well
Thank you. The binding affinity was slightly higher for SlMDH2, SlMDH6 and SlMDH9 for oxaloacetate over the malate, but it was the same for MDH11.
In my opinion there are no reasons to somehow say that from all the identified and characterized MDHs in tomato, the 11th one is somehow more involved in salt stress responses. Thus, the title has to be also adjusted accordingly.
Thank you. We agree with the reviewer that it is a tough decision to call out for SlMDH11. However, among the four highly expressed (SlMDH2, SlMDH6, SlMDH11, and SlMDH9), only SlMDH11 was co-localized with QTLs. We change the title as recommended.
Material and Methods
Line 334
BLAST is a tool not a verb.
Thank you. we corrected the text.
Line 335
“With the help of ProtParam Tool,” – please, add citation.
Thank you. we elaborated the text.
Line 343
“motifs were selected as 10 and the other parameters used were default” – instead, I suggest to write “motifs were selected as 10 and the other parameters were used as default”
Thank you. we modified the text.
Line 346
Not “4.3 Phylogenetic analysis and sequence alignment” but “4.3 Sequence alignment and phylogenetic analysis”
Thank you. we modified the text.
Line 350
Which MEGA was used – 6 or 7? (somewhere before you wrote 6).
Thank you. we used MEGA6 and we also modified the text.
Line 350-351
Try to combine it with the first sentence in this part of the text.
Thank you. we modified the text.
Line 355
The name of the program is PlantCARE. Please, cite it.
Thank you. we cited as suggested.
Line 358-365
This part – “Synteny and collinearity analysis” – should be placed before “4.3 Phylogenetic analysis and sequence alignment”. Part 4.2 should go after the Phylogenetic one.
Thank you. we modified the text as suggested.
Line 367-377
Cite the databases. I believe more information on treatments has to be written.
Thank you. we modified the text.
Line 381
The tomato name has to be editd.
Thank you. we modified the text.
Line 382
Seeds were sown and plant were grown at
Thank you. we modified the text.
Line 384
How old were these seedlings?
Thank you. Two weeks seedlings were used for treatment and samples were harvested after 3 and 6 hours.
Line 385
Please, write the composition of the solution.
Thank you. we modified the text.
Line 386-388
What was the volume of the original solution and then how much of the 250 mM solution was added?
Thank you. we modified the text.
Line 391
You do not show results for these timepoints.
We apologize for the inconvenience. We corrected the text.
Line 396
To the list in the ST2, please, add primers for actine.
Thank you. we added the primer.
Line 397
Please, add citation for the method used to calculate relative expression.
Thank you. We cited the method we used for expression analysis.
Line 404
Citations for the sources
Thank you. We cited the references.
References
Please, remember about writing Latin names of species in italics.
- I guess it is better to write as authors “Sato S,…. and the other authors (till the number of authors allowed by IJMS). The “x” is I guess not needed.
Thank you. Actually, there were more authors in this publication’s we checked the other papers also and got the same result as we used.
- I believe there are some mistakes in the citation.
Thank you. We corrected the references
- The paper that you cite is just a review. I suggest then to rather cite the program with the website of use, according to the standards of IJMS. Otherwise, I believe the review should be cited including following information: “J. Chem. Inf. Model. 2010, 50, 11, 2053. “ (in a way adjusted to the requirements of IJMS).
Thank you. We corrected the references

Reviewer 2 Report
In this manuscript, the authors identify 12 Malate dehydrogenase genes in tomato, Solanum lycopersicum. The authors then perform phylogenetic, gene expression, molecular docking, and QTL-colocalization analyses to characterize these genes and their products, finding that MDH11 can potentially be used as a candidate gene to explore mechanisms of salt tolerance in tomato.
The paper is, generally, well written, and the experiments and analyses well performed. I have only a few comments.
Again, generally, the paper is well written and easily readable. Early in the paper there are a few instances of, e.g., missing articles ("the", "a/an", etc). However, I find the Discussion section to contain quite a few more of these types of errors than the other parts of the paper. A specific example: p. 14, line 256: "MDH is a very active in plant cell and is required for numerous metabolic activities." The "a" should be removed and "cell" changed to "cells". These errors are minor but a bit distracting. Hence, I would suggest some minor editing by a native English speaker, if possible.
A few specific comments:
p.2, line 82: "The genome of the tomato is just about 950 megabytes in size ..." Should this be megabases?
p.3, line 111: "RT-qPCR results of MDH genes, in response to slat stress and ..." slat -> salt
Table 1: I may have just missed it, but after reading through the paper, it is still not clear to me how the subcellular localization was determined.
In the text, Figure 5 is called out before Figure 4; should they be switched?
Figure 7: I feel that the "zoomed out" images of the complete protein structures are too small, and there appears to be some inset text that is completely unreadable. The text should either be enlarged to be readable or removed.
Author Response
Reviewer
In this manuscript, the authors identify 12 Malate dehydrogenase genes in tomato, Solanum lycopersicum. The authors then perform phylogenetic, gene expression, molecular docking, and QTL-colocalization analyses to characterize these genes and their products, finding that MDH11 can potentially be used as a candidate gene to explore mechanisms of salt tolerance in tomato.
The paper is, generally, well written, and the experiments and analyses well performed. I have only a few comments.
Again, generally, the paper is well written and easily readable. Early in the paper there are a few instances of, e.g., missing articles ("the", "a/an", etc). However, I find the Discussion section to contain quite a few more of these types of errors than the other parts of the paper. A specific example: p. 14, line 256: "MDH is a very active in plant cell and is required for numerous metabolic activities." The "a" should be removed and "cell" changed to "cells". These errors are minor but a bit distracting. Hence, I would suggest some minor editing by a native English speaker, if possible.
We thank the reviewer for suggestions/comments that helped us a lot to improve the quality of the manuscript.
A few specific comments:
p.2, line 82: "The genome of the tomato is just about 950 megabytes in size ..." Should this be megabases?
Thank you. We corrected the text.
p.3, line 111: "RT-qPCR results of MDH genes, in response to slat stress and ..." slat -> salt
Thank you. We corrected the text.
Table 1: I may have just missed it, but after reading through the paper, it is still not clear to me how the subcellular localization was determined.
Thank you. The subcellular localization of each MDH protein was predicted using WoLF PSORT (http://www.genscript.com/wolf-psort.html).
In the text, Figure 5 is called out before Figure 4; should they be switched?
Thank you. We corrected the text.
Figure 7: I feel that the "zoomed out" images of the complete protein structures are too small, and there appears to be some inset text that is completely unreadable. The text should either be enlarged to be readable or removed.
Thank you. All the necessary information is given in table 2 and it is the best way to compare four proteins as well. Now we have made its better and added the information.

Round 2
Reviewer 1 Report
Reviewer
All my present comments are written in blue and marked in bold.
The manuscript entitled “Genomic Identification, Molecular Characterization and QTL Mapping of Malate Dehydrogenase Genes confers SlMDH11 is Potential Candidate for Salt Tolerance in Tomato” is an interesting work. The Authors identified and characterized tomato malate dehydrogenase (MDH) genes and encoded by them proteins known to be responsible for the reversible conversion of malate to oxaloacetate. The Authors presented results, which were obtained applying in silico tools (broad range) and by carrying out qRT-PCRs experiments.
Although the work is quite interesting, it needs further improvement. Comment and suggestions are written below.
We thank the reviewer for suggestions/comments that helped us a lot to improve the quality of the manuscript.
The manuscript has been improved. However, I still see elements/parts that have to be/could be improved. Please, find more information in the text below.
General comments:
1. Please, make sure to write gene names always in italics, including in figures (whenever possible; e.g. Figure 4) or when you write MDH gene family, MDH group, etc.
Thank you. We modified the text as suggested.
2. While applying the style with starting a word with a big letter, please, make sure where to use them and when not.
Thank you. We modified the text as suggested.
3. I strongly suggest to write the part Materials and Methods with more details.
Thank you. We elaborated our material and methods.
4. Improve English.
Thank you. We carefully read the manuscript.
5. Please, try to improve figures for the RNA seq data. They already look much better but I think if you keep “0” as white, it would make it not so confusing. Appropriate interpretation of the data presented in these figures is the major problem of the manuscript at this point, in my opinion.
Title
I would be very careful with using the term “OTL mapping” in the title as you did not perform a typical QTL mapping study.
I think “a Potential Candidate”.
Thank you. We changed the title.
I suggest to not write the title in a way that it implies characterization of MDH under salt stress as only expression studies were done under salt stress.
Affiliations
Please, try to edit affiliation part so that it is uniform.
Please mark the meaning of “+”.
Thank you. We edited the affiliation so that they are uniform.
Abstract
Line 25
After Solanum lycopersicum write L. or Mill.
Thank you. We modified the text as suggested.
The “L.” should not be written in italics.
Line 27
“The tomato MDH genes were split into five groups based on phylogenetic analysis, with genes that clustered together sharing the same subcellular position and having similar sequence lengths, gene structures, and conserved motifs.”
That is not fully correct as e.g.: genes in the group II localize to chloroplasts and glyoxysomes, in the group III to chloroplasts and cytosol, and so on.
Thank you. We modified the text.
Line 33
“at various stages” – here and in other places in the manuscript
Thank you. We modified the text.
Line 35
I would not come to this conclusion that “especially salt stress”. I suggest to remove it.
Thank you. We modified the text as suggested.
Line 36
There should not be “,” after “to abiotic stresses”.
Thank you. We modified the text as suggested.
1. Introduction:
Line 50
I would be very careful here. I suggest to remove “all”. In the publication that you cite and e.g. in [5] it states most of the bacteria species, not all. In general, I suggest to place here a more updated reference.
Line 56
I believe to write “The active site of MDH proteins is located in a cleft between the two domains, and each subunit has two distinct domains: a NAD/NADP-binding domain and a substrate-binding domain [4, 5].” is not clear. Thus, I suggest to first write about domains in each subdomain and then about the active site between these two domains.
Thank you. We modified the text as suggested.
Line 60-64
Citations
Thank you. We added the citations.
Line 67
Citation for each species
Thank you. We added the citations.
Line 68
I believe you meant 13, 13 and 25.
Thank you. Yes, it should be 13, 13, and 25 and modified the text.
It is always better to not start a sentence with a number.
Line 73
MDH2 not mDH2
Thank you. It was mitochondrial MDH2 and corrected the text.
Line 74-76
Since you write about the engagement of MDH proteins in plant responses to high salt concentration later, at the end of this page, the following information “The overexpression of NADP-74 MDH can enhance salt tolerance in Arabidopsis thaliana [12]. Likewise, apple cytosolic MDH can increase the cold and salt tolerance [13, 14].” could be shortened here. You could just state that MDH proteins were shown to be involved in plant responses to different abiotic stresses, e.g. cold, salinity, and leave the citations that you have at the moment. The more exact information could be added to the part about salt stress at the end of this page.
Thank you. We modified the text as suggested.
Line 76
“As a result” does not fit here. I suggest to write e.g. “In summary”, and not “over the evolutionary process of different species” but e. g. “over the course of evolution”.
Thank you. We modified the text as suggested.
Line 80
After Solanum lycopersicum write L. or Mill.
Tomato is not a model plant for the production of fleshy fruits but rather for studying fleshy fruits.
Thank you. We modified the text as suggested.
The “L.” should not be written in italics.
Line 82
I suggest to cite the size of the tomato gnome with some more updated source.
Thank you. We modified the text.
Line 83
I believe you wanted to write that tomato plants have a relatively short life cycle, not the genome of the species. Please, try to correct it.
Thank you. We modified the text.
Line 86
The citation [17] is the same as [15]. Moreover, if you want to write about tomato genome sequencing successes, I suggest to use the most updated sources (not from 2012).
Thank you. We are not focusing on tomato genome sequencing history, therefore, we modified the text.
Line 86-87
“Despite the fact that tomato production has been steadily increasing in recent years,” – please, add citation.
Thank you. We added the citation.
Line 92
Once you start to write about MDH, I suggest to start it with a new sentence.
Thank you. We modified the text as suggested.
Line 94-95
Just before writing the following sentence: “For instance, rice plastidial NAD-dependent MDH1 reduces the response to salt stress by modifying the vitamin B6 concentration of rice tissues [18].” you mention about the identification of possible gene that may confer resistance to abiotic stresses. The way you write is contradictory. I suggest to better write about genes that regulate plant responses to high salt concentration, not confer resistance. Then you can write that MDH proteins were found as both positive and negative regulators of plant responses to salt stress. Eventually, you could enumerate the examples, including the ones from lines 75-76.
Thank you. We modified the text as suggested.
Line 125
Please, write somewhere “poplar” as at this moment the meaning of the abbreviation “Pt” before the gene name is not known.
Line 132
Two times “crop”
Line 137
“At various” – please, correct it everywhere as asked before.
Line 139
Not “RT-qPCR result of MDH genes” but “RT-qPCR result of MDH gene expression”
Line 143
Not “of MDHs in regulating” but “the role of MDHs in regulating”
Results
Line 119-121
“The candidate MDH genes were identified through a systematic BLAST search against the tomato genome database with query sequences of G. arboretum and Arabidopsis.”
It does not correspond with what is written in Materials and Methods (lines 331-334):
“Solanum lycopersicon MDH family sequences were retrieved from Phytozome v12.1 [33]. Arabidopsis thaliana, Gossypium hirsutum and Oryza sativa sequences obtained from Phytozome v12.1 database, were used as query sequences.
Please, write in details the steps taken. Write also exactly what kind of sequences did you use as queries. I believe not all Arabidopsis thaliana, Gossypium hirsutum and Oryza sativa MDH sequences? Please, specify what type of sequences were used – amino acid, nucleotide (best in Materials and Methods section).
Thank you. We modified the text as suggested.
That part is still quite unclear, both here and especially in the section Materials and methods.
Here, you write “The candidate MDH genes were identified through a systematic BLAST search against the tomato genome database with nucleotide MDH query sequences of Gossypium hirsutum, Oryza sativa and Arabidopsis thalianaas queries”.
Thus, I understand that you used Gossypium hirsutum, Oryza sativa and Arabidopsis thaliana MDH gene nucleotide sequences in BLASTN to search homologous ones in tomato genome.
In Materials and methods you write “Solanum lycopersicon MDH family sequences were retrieved from Phytozome v12.1 [41]. Arabidopsis thaliana, Gossypium hirsutum and Oryza sativa sequences, obtained from the Phytozome v12.1 database, were used as queries in BLASTP [42] searches against Solanum Lycopersicon genome database.”
Here, you write that you used BLASTP, and at this point I am confused as in BLASTP you should use amino acid sequences to obtain amino acid sequences. Please, try to clarify it.
Write also exactly what kind of sequences did you use as queries. I believe not all Arabidopsis thaliana, Gossypium hirsutum and Oryza sativa MDH sequences? If all, please, write it.
Line155
You wrote “ProtParam (https://web.expasy.org/protam/) was utilized in order to characterize the physiological and biochemical parameters of the SlMDH proteins and the results are shown in Table 1.”
and in Material and methods you write:
“Finally, the physicochemical parameters of the full-length tomato MDH proteins were calculated by ExPASy (http://cn.expasy.org/tools) [45].” Please, try to make it uniform.
I suggest to add information on identifiers for amino acid sequences used in this study (I did not find the supplementary file).
Line 127
Instead of “investigations” I suggest to write “parameters”.
Thank you. We modified the text as suggested.
Line 131-134
“The number of amino acids contained in SlMDH proteins 131 varied in length, going anywhere from 298 to 467. With a mean value of 38.99 132 KDa, the molecular weights of MDH proteins ranged anywhere from 31.81 133 to 51.66 KDa.” – it is a repetition of what is written in the lines before in your manuscript.
Thank you. We modified the text.
Line 137
Please, write in Materials and Methods the program that you used to assess protein localization.
Thank you. We added the information in material and methods.
Line 1– 45
This part needs quite a lot of improvement. It has to be better explained and written.
Thank you. We modified the text as suggested.
Line 15
Not ”and the results showed that there are” but “and the results showed that there were”
Line 16
“duplication events, i.e., one between SlMDH4/SlMDH11 and the other between SlMDH6/SlMDH12” – it is not clear which genes are duplicates from the way you write it and there is no figure citation. If SlMDH4 and SlMDH11 are duplicates then the sentence following this one is not correct as they are in the same group.
Line 13-14
After the following sentence: “In addition, during the process of tomato domestication, both whole-genome duplication (WGD) and segmental duplication occurred, both of which had an essential part in the expansion of the apple gene family.” please, write citation.
Could you also be more specific with the apple gene family (which genes?), and why do you write about apple. That has to be explained.
Thank you. We modified the text.
Line 16
I believe “the growth” in “the growth of SlMDH genes” is not the best word to use.
Thank you. We modified the text.
Line 19
“The Ka/Ks ratio is an important measure for determining the number of replication events as well as the selection forces [21].” – do you mean here duplication events?
Thank you. Yes, it is duplication of events and corrected the text.
Line 20-22
“We calculated evolutionary rates and selective pressure by using the Ka (nonsynonymous), Ks (synonymous),” – try to explain it better, adding some information about substitution per site.
Thank you. We modified the text as suggested.
Line 24-26
“Our goal was to determine whether or not a SlMDH gene is under different selection pressures and has different evolutionary rates than its duplicated genes [21].” - it is a repetition of what is written in the lines before in your manuscript.
Thank you. We removed the repetitions in the text as suggested.
Line 26-28
“As a result, we studied the selective pressures that are exerted on duplicated MDH genes; the values of each duplicated gene pair are presented.” – it is not clear where are the values presented.
Thank you. We mentioned Table S2 in the text.
Line 30
What do you mean by writing more pure forms?
Thank you. We modified the text.
Line 31-32
Sentence not grammatically correct.
Thank you. We modified the text.
Line 35-36
“the evolutionary mechanism of SlMDH members.” – it is not correct. SlMDH members do not have an evolutionary mechanism.
Thank you. We modified the text.
Line 36-43
All information about particular genes should be placed in the figure 1A and 1B. Otherwise, it is hard to analyze such data.
Thank you. We have written the information in the text very clear, but it’s hard and create overlapping of genes. Therefore, we used the figures as such obtained by the software.
Line 53 (Figure 1 description)
“show” instead of “showed”
Thank you. We corrected the text.
Line 75
I would not say that these results reveled common ancestors of these genes.
Line 76
I think there are no 3 corresponding A. thaliana genes – only 1 or 2.
Line 55-80
I suggest to create e. g. a supplementary file with information on accession numbers for all sequences used to create the phylogenetic tree. It also has to be clearly stated in the Material and Methods section how did you obtain these sequences.
Thank you. We added the supplementary file.
I did not get it along with the supplementary files. I think it did not get uploaded.
Line 66-69
“and it was found that group V had a greater number of members than the other three groups combined. In addition to this, it was found that the SlMDHs have a greater evolutionary link with the AtMDHs and the GaMDHs within each group.”
I suggest to remove that part. First element is not important, and the last sentence is not correct. I suggest just to explain into which group fall particular tomato MDH protein.
Thank you. We modified the text as suggested.
Line 103
I suggest to remove the information about bootstrap values.
Line 108-113
I do not think that your conclusion is correct.
Line 72
The figure 2 is not split into subfigures. Thus, I recommend to remove the letter “A” from the figure description part.
Thank you. We corrected the text.
Line 120
From my understanding, you carried out the phylogenetic analyses using amino acid sequences, as stated at the beginning of this part of results. Thus, I suggest to not write here MDH genes but proteins.
General comment – since phylogenetic analyses and motif analyses were done using protein sequences (from my understanding) that has to be somehow made clear in the title of this part of results.
Line 77
You could write not just about colors but also about figures that were chosen for particular species.
Thank you. We modified the text as suggested.
Line 84-88
Since here it is not easy to follow, which SlMDH proteins fall into particular group, I suggest to write names of SlMDH in brackets when writing about groups. Moreover, earlier to describe these groups you used Roman letters, and here Arabic. Please, try to make it more uniform.
Thank you. We corrected the text as suggested.
Line 133
What are these motifs? Please, state in brackets.
Line 136-137
This information is not correct.
Line 140-146
The groups that you write here about do not correspond to the ones that you obtained using phylogenetic tree. Your group I here is actually group III, and so on. That has to be corrected.
Line 85
“while motif 3 and motif 7 are absent in Group 5” – that is not fully true as SlMDH1 has motif 3.
Thank you. We corrected the text.
Line 87
„motif 10 is only present in Group 4 and Group 5,” – again, not correct. This motif is present only in group 5.
Thank you. We corrected the text.
Line 98
Figure 3, not 5.
Thank you. We corrected the text.
Line 99
“in SlMDH genes ranged from 0 to 13, with SlMDH11 having the most introns,” – I guess both 11 and 4?
Thank you. We corrected the text.
Line 104-116
I suggest to write more information about the particular cis-regulatory elements found in the promoter regions.
Thank you. We modified the text.
Line 105
Instead of writing “in the upstream 2 kb promoter regions of SlMDH genes” I suggest to write upstream of the transcription start site.
Thank you. We modified the text.
Line 108
Are you sure about CAT boxes? Aren’t they CAAT boxes? (here and in the figure).
There are no TATA boxes presented in the figure.
Thank you. We modified the text, as CAAT and TATA box are commonly present in each promoter. Therefore, we did not mention them in the figure. However, we found the CAT-box and mentioned in the figure.
Line 169-170
Please, write after this sentence in brackets “result not shown” to not bring any confusion.
Line 162-181
I suggest to still write bit more about particular motifs found. For the reader it has to be stated that motifs, which can be put into this and that group/responsible for this and that, were found. For example, what is the function of MBS motif, CAT-box, and so on. It should not be left like that.
Line 118
All promoters have some regulatory sequences. The sentence has to be rewritten.
Thank you. We modified the text.
I suggest to write: “Cis-regulatory motifs found in promoters of….” And then the second sentence.
Line 121
“tissues” not “tissue”
Thank you. We corrected the text.
Line 121-134
First, I suggest to place the SF2 in the main text and to describe it in details, especially particular stages as it is not fully clear, e,g, what is the age of flowers (days after pollination), what means Breaker +10, etc.
Thank you. We brought the SF2 into the main text (Now Fig 5) and corrected the text. The detail of different stages are given in the figure legends.
Please, correct also the description in the text as not all the information that you write about the results is correct in relation to what is presented in the SF2.
Thank you. We corrected the text.
Line 193
To make it easier for the reader, please, write everywhere the same name for the stages analyzed, and in the same order – both in the text here and in the one describing the figure.
Line 198
It rather suggests functions of these genes at various plant developmental stages and in different tissues, not necessarily different ones.
Line 201-202
Since white does not mean “0” according to the color scale applied, the blue for MDH11 expression in roots may actually mean 0 (?)
Line 202-204
I believe to write here about intermediate expression is not correct. Maybe in relation to other expression it could be described as intermediate. Try to rewrite it.
Line 215
“in the middle” – I believe it is not needed here.
Line 136-148
Either in the section Materials and Methods or here, describe in more details the treatments.
Please, correct the description for the results presented in the figure 5 as not all the information that you write about these results is true. There is quite substantial number of mistakes. To avoid such mistakes, I suggest to not write about the results in a too general way.
Thank you. We modified the text.
List the 7 genes that you write about,
Thank you, We modified the text.
Please, refer to the Recovery phase in the figure 5. What does it mean? Try to describe it more and explain these results as well.
Thank you, We modified the text and added the information on the recovery phase.
Line 153-155
Please, describe all abbreviations used in the figure. Moreover, I strongly suggest to make the scale more uniform, e.g. from -1 to 3 in all figures, and for “-“ results use the blue color, for “0” white, and for the “+” red. At this point it is quite confusing. I think some confusion in describing these results may results from this not uniform color code.
Thank you, We modified the figure according to your suggestions.
I like the improvements in the figure. Now it seems more clear. However, still, “0” as white would make it much better.
Please, describe all abbreviations used in the figure.
Line 222-224
Actually, this is not correct. It seems like the expression of these four genes is relatively high independent of stress application as you see that it is high both at “0” time-points/in control samples and for treated ones. For MDH2 or MDH11 it even seems to drop along with the time of treatment. That has to be very well though through and written correctly. Such interpretation changes also slightly the idea of your work. You have to be very careful when writing the abstract, and conclusions.
Line 225-226
MDH1 – probably yes, but not for MDH7 (it is probably below 0 but does not decrease along with the stress). Keep also in mind that for salt stress you have time-points both for the control and treated samples. That also have to be correctly interpreted.
Line 227-229
The interpretation is not fully correct.
Line 229-230
Repetition of what was written earlier.
Not “from RNA-sequencing” but e.g. “based on RNA-sequencing data” or “using RNA-sequencing data”
Thank you, We modified the text.
Line 159
Not “by exposing tomato plants to salt stress” but “on plants exposed to salt stress”
Thank you, We modified the text.
Line 257
Maybe not “interval” but “0, 3, and 6 hours after treatment” (here and in figure description).
Line 174
Statistical analysis of difference significance is needed for the gene expression studies.
Thank you, We applied the T-test.
Line 273
How many biological replicates have you used?
Line 183-184
There is no information on residues in Figure 7. It would be actually good to add such information.
Thank you. The interacting residues and binding information are already given in table 2. In addition, it is a lot easier to compare the four proteins in table 2.
Line 280
Considering the changes in interpretation of the RNA-seq expression data I think “highly expressed in all the abiotic stresses” does not fit here.
Line 195
Correct the result for oxaloacetate
Thank you. we corrected the text.
Line 194-195
“However, the NADH molecule showed the strongest binding with all the SlMDHs proteins” – not correct, for SlMDH2 – the same for NAD and NADH
Thank you. we corrected the text.
Line 204
“2, 4 and 3 hydrogen bonds” not “2, 4 and 2 hydrogen bonds”
Thank you. we corrected the text.
Line 219
What about NADP? How do you know which MDH has NAD+ or NADP+ cofactor?
According to the malate valve hypothesis, NADPH is generated in the chloroplast through the electron transport chain during the day but does not readily diffuse out of the chloroplast. NADP-MDH reduces oxaloacetate to malate by oxidizing NADPH to NADP, and the malate is shuttled to the cytosol in exchange for oxaloacetate via the dicarboxylate transporter. In the cytosol, malate is oxidized back to oxaloacetate by the cytosolic MDHs, regenerating a reducing equivalent in the form of NADH in that compartment. So when the MDH is located in chloroplast it has NADP+ cofactor, while cytosolic MDH has NAD+ cofactor.
I suggest to explain in the publication why you did not carry out these analyses with NADP+ for the MDH11, which likely localizes in chloroplasts.
It is known from kinetic studies that the malate to oxaloacetate reaction is an ordered reaction with NAD/NADH binding first, followed by the dicarboxylic acid substrate (Silverstein et al. 1969). It would mean that the docking for malate has to be done for the MDH molecule with the cofactor. Is it possible to do it?
This is possible but it will need a lot of time as this is a very detailed study and requires a lot of computational resources. However, this is not the center of the current study, but we thank the reviewer for a nice suggestion.
Please, describe the meaning for all abbreviations used in the table.
Thank you. we modified the text by adding the abbreviations for each.
Line 222-225
What are these ligands exactly with particular SlMDH? Please, add that information.
I suggest to write just “blue” as there are different blue colors for proteins in figure 7.
Thank you. we modified the text.
Line 228-237
Please, cite all the QTL studies that were used by you to mark QTLs in Figure 8. I also suggest to at least write their names while describing them in the text.
Thank you. we cited the references and modified the text.
Line 351
Please, correct names of QTLs so they are as in the figure.
Line 337-358
It seems like more MDH genes co-localize with the chosen QTLs. It is then not clear while you still claim (including abstract) that especially MDH11 co-localizes with salt stress-related QTL. That has to be clearly explained. You should make clear statement which of the described characteristics are salt stress-related.
Discussion
Line 254
“MDHs are very…” not “MDH is a very”
Thank you. we modified the text.
Further part of this sentence has to be also corrected (now line 367).
Line 373
The sentence is missing a part.
Line 378-380
That is not really true.
Line 383
When protein name-no italics.
Line 395
That is not your group I but III
Line 414
Gene names – italics
Line 418
I would not come to conclusion that each gene exhibit a distinct expression.
Line 254-263
There are some unnecessary repetitions, and quite some grammatic mistakes.
Thank you. we modified the text.
Citation [24] is not about MDHs.
Thank you. we corrected this mistake.
Line 268
These genes are not localized in chloroplasts but their products.
Four groups not two.
Thank you. we removed the text as it was not fitting properly in the context of the paragraph here.
Line 270
I am already confused whether you analyzed the motifs using gene sequences or amino acid ones. Here you write genes, in lines 80-82 protein sequences, and then in Material and Methods (4.2) – genes. That has to be clarified.
We apologize for the inconvenience. We used the amino acid sequence for motifs, and we mentioned it in the text as well.
Line 274
You did not show or write such results. I suggest to transfer this information to the results part.
Thank you. we transfer this information in the text.
Line 296
Please, write names of the duplicated genes.
Thank you. we modified the text.
Line 302-303
There were no 3 groups for the RNA-seq data.
Thank you. we modified the text.
Line 304-307
You have to be mor specific. That does not apply to all the genes studied.
We apologize for the inconvenience. We removed the sentences it was misleading and did not properly present the facts.
Line 307-308
There is no previous information (and in Materials and Methods) that the expression was analyzed in roots. All this has to be more explained.
Thank you. we modified the text.
I suggest to mention that also in part Results.
Line 310
“especially the SlMDH2, and SlMDH11 gene at 3hr” – it is not correct – 2 and 12.
Thank you. we corrected the text.
Line 425
The word “respectively” is missing. At this point, it is understood as both gene were highly expressed at both time-points.
Line 437
Please, take into account the proper interpretation of the data.
Line 315-317
This information is not correct.
Thank you. we corrected the text.
Line 324-325
“Our docking results showed that binding affinity of SlMDH11 is similar between oxaloacetate and malate,” – it was similar for other analyzed SlMDHs as well
Thank you. The binding affinity was slightly higher for SlMDH2, SlMDH6 and SlMDH9 for oxaloacetate over the malate, but it was the same for MDH11.
In my opinion there are no reasons to somehow say that from all the identified and characterized MDHs in tomato, the 11th one is somehow more involved in salt stress responses. Thus, the title has to be also adjusted accordingly.
Thank you. We agree with the reviewer that it is a tough decision to call out for SlMDH11. However, among the four highly expressed (SlMDH2, SlMDH6, SlMDH11, and SlMDH9), only SlMDH11 was co-localized with QTLs. We change the title as recommended.
Material and Methods
Line 447
What kind of sequences – nucleotide, amino acid?
Line 451
In InterProScan I believe you used amino acid sequences not gene ones?
Line 487
You already stated that motif analysis was carried out with amino acid sequences. Thus, it can not be written as “motif analysis of MDH genes”.
Line 334
BLAST is a tool not a verb.
Thank you. we corrected the text.
Line 335
“With the help of ProtParam Tool,” – please, add citation.
Thank you. we elaborated the text.
Line 343
“motifs were selected as 10 and the other parameters used were default” – instead, I suggest to write “motifs were selected as 10 and the other parameters were used as default”
Thank you. we modified the text.
Line 346
Not “4.3 Phylogenetic analysis and sequence alignment” but “4.3 Sequence alignment and phylogenetic analysis”
Thank you. we modified the text.
Line 350
Which MEGA was used – 6 or 7? (somewhere before you wrote 6).
Thank you. we used MEGA6 and we also modified the text.
Line 350-351
Try to combine it with the first sentence in this part of the text.
Thank you. we modified the text.
Line 355
The name of the program is PlantCARE. Please, cite it.
Thank you. we cited as suggested.
Line 358-365
This part – “Synteny and collinearity analysis” – should be placed before “4.3 Phylogenetic analysis and sequence alignment”. Part 4.2 should go after the Phylogenetic one.
Thank you. we modified the text as suggested.
Line 367-377
Cite the databases. I believe more information on treatments has to be written.
Thank you. we modified the text.
Line 509-513
There are some repetitions.
Line 381
The tomato name has to be editd.
Thank you. we modified the text.
Line 382
Seeds were sown and plant were grown at
Thank you. we modified the text.
Line 384
How old were these seedlings?
Thank you. Two weeks seedlings were used for treatment and samples were harvested after 3 and 6 hours.
How old were these seedlings when transferred to the Hoagland solution?
Line 385
Please, write the composition of the solution.
Thank you. we modified the text.
Line 386-388
What was the volume of the original solution and then how much of the 250 mM solution was added?
Thank you. we modified the text.
You mean you added so much of the NaCl solution so the medium was eventually 250 mM NaCl (we do not write mmol/L)?
Line 391
You do not show results for these timepoints.
We apologize for the inconvenience. We corrected the text.
Line 396
To the list in the ST2, please, add primers for actine.
Thank you. we added the primer.
Line 397
Please, add citation for the method used to calculate relative expression.
Thank you. We cited the method we used for expression analysis.
In the Results part you write that you applied t-test. I do not understand then why you write about ANOVA.
Line 404
Citations for the sources
Thank you. We cited the references.
References
Please, remember about writing Latin names of species in italics.
15. I guess it is better to write as authors “Sato S,…. and the other authors (till the number of authors allowed by IJMS). The “x” is I guess not needed.
Thank you. Actually, there were more authors in this publication’s we checked the other papers also and got the same result as we used.
34. I believe there are some mistakes in the citation.
Thank you. We corrected the references
46. The paper that you cite is just a review. I suggest then to rather cite the program with the website of use, according to the standards of IJMS. Otherwise, I believe the review should be cited including following information: “J. Chem. Inf. Model. 2010, 50, 11, 2053. “ (in a way adjusted to the requirements of IJMS).
Thank you. We corrected the references

Author Response
We really appreciate your excellent, very clear, and critical suggestions, comments as well recommendations to improve our paper. Thank you very much.
Reviewer
All my present comments are written in blue and marked in bold.
All the answers for 2nd revision were written in red bold color.
The manuscript entitled “Genomic Identification, Molecular Characterization and QTL Mapping of Malate Dehydrogenase Genes confers SlMDH11 is Potential Candidate for Salt Tolerance in Tomato” is an interesting work. The Authors identified and characterized tomato malate dehydrogenase (MDH) genes and encoded by them proteins known to be responsible for the reversible conversion of malate to oxaloacetate. The Authors presented results, which were obtained applying in silico tools (broad range) and by carrying out qRT-PCRs experiments.
Although the work is quite interesting, it needs further improvement. Comment and suggestions are written below.
We thank the reviewer for suggestions/comments that helped us a lot to improve the quality of the manuscript.
The manuscript has been improved. However, I still see elements/parts that have to be/could be improved. Please, find more information in the text below.
Note: We really appreciate your excellent, very clear, and critical suggestions, comments as well recommendations to improve our paper. Thank you very much.
General comments:
- Please, make sure to write gene names always in italics, including in figures (whenever possible; e.g. Figure 4) or when you write MDHgene family, MDH group, etc.
Thank you. We modified the text as suggested.
- While applying the style with starting a word with a big letter, please, make sure where to use them and when not.
Thank you. We modified the text as suggested.
- I strongly suggest to write the part Materials and Methods with more details.
Thank you. We elaborated our material and methods.
- Improve English.
Thank you. We carefully read the manuscript.
- Please, try to improve figures for the RNA seq data. They already look much better but I think if you keep “0” as white, it would make it not so confusing. Appropriate interpretation of the data presented in these figures is the major problem of the manuscript at this point, in my opinion.
Thank you. We have improved the figures according to your kind suggestions.
Title
I would be very careful with using the term “OTL mapping” in the title as you did not perform a typical QTL mapping study.
I think “a Potential Candidate”.
Thank you. We changed the title.
I suggest to not write the title in a way that it implies characterization of MDH under salt stress as only expression studies were done under salt stress.
Thank you. We have modified the title.
Affiliations
Please, try to edit affiliation part so that it is uniform.
Please mark the meaning of “+”.
Thank you. We edited the affiliation so that they are uniform.
Abstract
Line 25
After Solanum lycopersicum write L. or Mill.
Thank you. We modified the text as suggested.
The “L.” should not be written in italics.
Thank you. The text is modified now.
Line 27
“The tomato MDH genes were split into five groups based on phylogenetic analysis, with genes that clustered together sharing the same subcellular position and having similar sequence lengths, gene structures, and conserved motifs.”
That is not fully correct as e.g.: genes in the group II localize to chloroplasts and glyoxysomes, in the group III to chloroplasts and cytosol, and so on.
Thank you. We modified the text.
Line 33
“at various stages” – here and in other places in the manuscript
Thank you. We modified the text.
Line 35
I would not come to this conclusion that “especially salt stress”. I suggest to remove it.
Thank you. We modified the text as suggested.
Line 36
There should not be “,” after “to abiotic stresses”.
Thank you. We modified the text as suggested.
- Introduction:
Line 50
I would be very careful here. I suggest to remove “all”. In the publication that you cite and e.g. in [5] it states most of the bacteria species, not all. In general, I suggest to place here a more updated reference.
Line 56
I believe to write “The active site of MDH proteins is located in a cleft between the two domains, and each subunit has two distinct domains: a NAD/NADP-binding domain and a substrate-binding domain [4, 5].” is not clear. Thus, I suggest to first write about domains in each subdomain and then about the active site between these two domains.
Thank you. We modified the text as suggested.
Line 60-64
Citations
Thank you. We added the citations.
Line 67
Citation for each species
Thank you. We added the citations.
Line 68
I believe you meant 13, 13 and 25.
Thank you. Yes, it should be 13, 13, and 25 and modified the text.
It is always better to not start a sentence with a number.
Thank you. The text is modified.
Line 73
MDH2 not mDH2
Thank you. It was mitochondrial MDH2 and corrected the text.
Line 74-76
Since you write about the engagement of MDH proteins in plant responses to high salt concentration later, at the end of this page, the following information “The overexpression of NADP-74 MDH can enhance salt tolerance in Arabidopsis thaliana [12]. Likewise, apple cytosolic MDH can increase the cold and salt tolerance [13, 14].” could be shortened here. You could just state that MDH proteins were shown to be involved in plant responses to different abiotic stresses, e.g. cold, salinity, and leave the citations that you have at the moment. The more exact information could be added to the part about salt stress at the end of this page.
Thank you. We modified the text as suggested.
Line 76
“As a result” does not fit here. I suggest to write e.g. “In summary”, and not “over the evolutionary process of different species” but e. g. “over the course of evolution”.
Thank you. We modified the text as suggested.
Line 80
After Solanum lycopersicum write L. or Mill.
Tomato is not a model plant for the production of fleshy fruits but rather for studying fleshy fruits.
Thank you. We modified the text as suggested.
The “L.” should not be written in italics.
Thank you. The text is modified now.
Line 82
I suggest to cite the size of the tomato gnome with some more updated source.
Thank you. We modified the text.
Line 83
I believe you wanted to write that tomato plants have a relatively short life cycle, not the genome of the species. Please, try to correct it.
Thank you. We modified the text.
Line 86
The citation [17] is the same as [15]. Moreover, if you want to write about tomato genome sequencing successes, I suggest to use the most updated sources (not from 2012).
Thank you. We are not focusing on tomato genome sequencing history, therefore, we modified the text.
Line 86-87
“Despite the fact that tomato production has been steadily increasing in recent years,” – please, add citation.
Thank you. We added the citation.
Line 92
Once you start to write about MDH, I suggest to start it with a new sentence.
Thank you. We modified the text as suggested.
Line 94-95
Just before writing the following sentence: “For instance, rice plastidial NAD-dependent MDH1 reduces the response to salt stress by modifying the vitamin B6 concentration of rice tissues [18].” you mention about the identification of possible gene that may confer resistance to abiotic stresses. The way you write is contradictory. I suggest to better write about genes that regulate plant responses to high salt concentration, not confer resistance. Then you can write that MDH proteins were found as both positive and negative regulators of plant responses to salt stress. Eventually, you could enumerate the examples, including the ones from lines 75-76.
Thank you. We modified the text as suggested.
Line 125
Please, write somewhere “poplar” as at this moment the meaning of the abbreviation “Pt” before the gene name is not known.
Thank you. We modified the text as suggested.
Line 132
Two times “crop”
Thank you. We modified the text as suggested.
Line 137
“At various” – please, correct it everywhere as asked before.
Thank you. We modified the text as suggested.
Line 139
Not “RT-qPCR result of MDH genes” but “RT-qPCR result of MDH gene expression”
Thank you. We modified the text as suggested.
Line 143
Not “of MDHs in regulating” but “the role of MDHs in regulating”
Thank you. We modified the text as suggested.
Results
Line 119-121
“The candidate MDH genes were identified through a systematic BLAST search against the tomato genome database with query sequences of G. arboretum and Arabidopsis.”
It does not correspond with what is written in Materials and Methods (lines 331-334):
“Solanum lycopersicon MDH family sequences were retrieved from Phytozome v12.1 [33]. Arabidopsis thaliana, Gossypium hirsutum and Oryza sativa sequences obtained from Phytozome v12.1 database, were used as query sequences.
Please, write in details the steps taken. Write also exactly what kind of sequences did you use as queries. I believe not all Arabidopsis thaliana, Gossypium hirsutum and Oryza sativa MDH sequences? Please, specify what type of sequences were used – amino acid, nucleotide (best in Materials and Methods section).
Thank you. We modified the text as suggested.
That part is still quite unclear, both here and especially in the section Materials and methods.
Here, you write “The candidate MDH genes were identified through a systematic BLAST search against the tomato genome database with nucleotide MDH query sequences of Gossypium hirsutum, Oryza sativa and Arabidopsis thalianaas queries”.
Thus, I understand that you used Gossypium hirsutum, Oryza sativa and Arabidopsis thaliana MDH gene nucleotide sequences in BLASTN to search homologous ones in tomato genome.
In Materials and methods you write “Solanum lycopersicon MDH family sequences were retrieved from Phytozome v12.1 [41]. Arabidopsis thaliana, Gossypium hirsutum and Oryza sativa sequences, obtained from the Phytozome v12.1 database, were used as queries in BLASTP [42] searches against Solanum Lycopersicon genome database.”
Here, you write that you used BLASTP, and at this point I am confused as in BLASTP you should use amino acid sequences to obtain amino acid sequences. Please, try to clarify it.
Write also exactly what kind of sequences did you use as queries. I believe not all Arabidopsis thaliana, Gossypium hirsutum and Oryza sativa MDH sequences? If all, please, write it.
Thanks a lot for you critical and valuable suggestions. We have used the amino acid sequence to search the MDHs genes. We modified the text as suggested in both places (material & methods, and results parts). We used three species sequences and their sequence are present in the supplementary file.
Line155
You wrote “ProtParam (https://web.expasy.org/protam/) was utilized in order to characterize the physiological and biochemical parameters of the SlMDH proteins and the results are shown in Table 1.”
and in Material and methods you write:
“Finally, the physicochemical parameters of the full-length tomato MDH proteins were calculated by ExPASy (http://cn.expasy.org/tools) [45].” Please, try to make it uniform.
Thank you. We modified the text as suggested and its uniform now.
I suggest to add information on identifiers for amino acid sequences used in this study (I did not find the supplementary file).
We used three species amino acid sequences and their sequence are presented in the supplementary file Table S2.
Line 127
Instead of “investigations” I suggest to write “parameters”.
Thank you. We modified the text as suggested.
Line 131-134
“The number of amino acids contained in SlMDH proteins 131 varied in length, going anywhere from 298 to 467. With a mean value of 38.99 132 KDa, the molecular weights of MDH proteins ranged anywhere from 31.81 133 to 51.66 KDa.” – it is a repetition of what is written in the lines before in your manuscript.
Thank you. We modified the text.
Line 137
Please, write in Materials and Methods the program that you used to assess protein localization.
Thank you. We added the information in material and methods.
Line 1– 45
This part needs quite a lot of improvement. It has to be better explained and written.
Thank you. We modified the text as suggested.
Line 15
Not ”and the results showed that there are” but “and the results showed that there were”
Thank you. We modified the text as suggested
Line 16
“duplication events, i.e., one between SlMDH4/SlMDH11 and the other between SlMDH6/SlMDH12” – it is not clear which genes are duplicates from the way you write it and there is no figure citation. If SlMDH4 and SlMDH11 are duplicates then the sentence following this one is not correct as they are in the same group.
Thank you. We modified the text as suggested. Secondly, we used TBtool to find the duplications events Table S1. According to the results they were duplicated.
Line 13-14
After the following sentence: “In addition, during the process of tomato domestication, both whole-genome duplication (WGD) and segmental duplication occurred, both of which had an essential part in the expansion of the apple gene family.” please, write citation.
Could you also be more specific with the apple gene family (which genes?), and why do you write about apple. That has to be explained.
Thank you. We modified the text.
Line 16
I believe “the growth” in “the growth of SlMDH genes” is not the best word to use.
Thank you. We modified the text.
Line 19
“The Ka/Ks ratio is an important measure for determining the number of replication events as well as the selection forces [21].” – do you mean here duplication events?
Thank you. Yes, it is duplication of events and corrected the text.
Line 20-22
“We calculated evolutionary rates and selective pressure by using the Ka (nonsynonymous), Ks (synonymous),” – try to explain it better, adding some information about substitution per site.
Thank you. We modified the text as suggested.
Line 24-26
“Our goal was to determine whether or not a SlMDH gene is under different selection pressures and has different evolutionary rates than its duplicated genes [21].” - it is a repetition of what is written in the lines before in your manuscript.
Thank you. We removed the repetitions in the text as suggested.
Line 26-28
“As a result, we studied the selective pressures that are exerted on duplicated MDH genes; the values of each duplicated gene pair are presented.” – it is not clear where are the values presented.
Thank you. We mentioned Table S2 in the text.
Line 30
What do you mean by writing more pure forms?
Thank you. We modified the text.
Line 31-32
Sentence not grammatically correct.
Thank you. We modified the text.
Line 35-36
“the evolutionary mechanism of SlMDH members.” – it is not correct. SlMDH members do not have an evolutionary mechanism.
Thank you. We modified the text.
Line 36-43
All information about particular genes should be placed in the figure 1A and 1B. Otherwise, it is hard to analyze such data.
Thank you. We have written the information in the text very clear, but it’s hard and create overlapping of genes. Therefore, we used the figures as such obtained by the software.
Line 53 (Figure 1 description)
“show” instead of “showed”
Thank you. We corrected the text.
Line 75
I would not say that these results reveled common ancestors of these genes.
Thank you. We modified the text as suggested.
Line 76
I think there are no 3 corresponding A. thaliana genes – only 1 or 2.
Thank you. We modified the text as suggested.
Line 55-80
I suggest to create e. g. a supplementary file with information on accession numbers for all sequences used to create the phylogenetic tree. It also has to be clearly stated in the Material and Methods section how did you obtain these sequences.
Thank you. We added the supplementary file.
I did not get it along with the supplementary files. I think it did not get uploaded.
We apologize for mistake. Yes, it was not uploaded last time. Now its uploaded now.
Line 66-69
“and it was found that group V had a greater number of members than the other three groups combined. In addition to this, it was found that the SlMDHs have a greater evolutionary link with the AtMDHs and the GaMDHs within each group.”
I suggest to remove that part. First element is not important, and the last sentence is not correct. I suggest just to explain into which group fall particular tomato MDH protein.
Thank you. We modified the text as suggested.
Line 103
I suggest to remove the information about bootstrap values.
Thank you. We modified the text as suggested
Line 108-113
I do not think that your conclusion is correct.
Thank you. We modified the text as suggested
Line 72
The figure 2 is not split into subfigures. Thus, I recommend to remove the letter “A” from the figure description part.
Thank you. We corrected the text.
Line 120
From my understanding, you carried out the phylogenetic analyses using amino acid sequences, as stated at the beginning of this part of results. Thus, I suggest to not write here MDH genes but proteins.
Thank you. We modified the text as suggested
General comment – since phylogenetic analyses and motif analyses were done using protein sequences (from my understanding) that has to be somehow made clear in the title of this part of results.
Thank you. We modified the text as suggested
Line 77
You could write not just about colors but also about figures that were chosen for particular species.
Thank you. We modified the text as suggested.
Line 84-88
Since here it is not easy to follow, which SlMDH proteins fall into particular group, I suggest to write names of SlMDH in brackets when writing about groups. Moreover, earlier to describe these groups you used Roman letters, and here Arabic. Please, try to make it more uniform.
Thank you. We corrected the text as suggested.
Line 133
What are these motifs? Please, state in brackets.
Thank you. We have added some information of different motifs regarding their functions and modified the text as suggested.
Line 136-137
This information is not correct.
Thank you. We modified the and corrected it.
Line 140-146
The groups that you write here about do not correspond to the ones that you obtained using phylogenetic tree. Your group I here is actually group III, and so on. That has to be corrected.
Thank you. We modified the text as suggested
Line 85
“while motif 3 and motif 7 are absent in Group 5” – that is not fully true as SlMDH1 has motif 3.
Thank you. We corrected the text.
Line 87
„motif 10 is only present in Group 4 and Group 5,” – again, not correct. This motif is present only in group 5.
Thank you. We corrected the text.
Line 98
Figure 3, not 5.
Thank you. We corrected the text.
Line 99
“in SlMDH genes ranged from 0 to 13, with SlMDH11 having the most introns,” – I guess both 11 and 4?
Thank you. We corrected the text.
Line 104-116
I suggest to write more information about the particular cis-regulatory elements found in the promoter regions.
Thank you. We modified the text.
Line 105
Instead of writing “in the upstream 2 kb promoter regions of SlMDH genes” I suggest to write upstream of the transcription start site.
Thank you. We modified the text.
Line 108
Are you sure about CAT boxes? Aren’t they CAAT boxes? (here and in the figure).
There are no TATA boxes presented in the figure.
Thank you. We modified the text, as CAAT and TATA box are commonly present in each promoter. Therefore, we did not mention them in the figure. However, we found the CAT-box and mentioned in the figure.
Line 169-170
Please, write after this sentence in brackets “result not shown” to not bring any confusion.
Thank you. We modified the text as suggested
Line 162-181
I suggest to still write bit more about particular motifs found. For the reader it has to be stated that motifs, which can be put into this and that group/responsible for this and that, were found. For example, what is the function of MBS motif, CAT-box, and so on. It should not be left like that.
Thank you. We have added little more information of cis-elements and modified the text as suggested.
Line 118
All promoters have some regulatory sequences. The sentence has to be rewritten.
Thank you. We modified the text.
I suggest to write: “Cis-regulatory motifs found in promoters of….” And then the second sentence.
Thank you. We modified the text as suggested
Line 121
“tissues” not “tissue”
Thank you. We corrected the text.
Line 121-134
First, I suggest to place the SF2 in the main text and to describe it in details, especially particular stages as it is not fully clear, e,g, what is the age of flowers (days after pollination), what means Breaker +10, etc.
Thank you. We brought the SF2 into the main text (Now Fig 5) and corrected the text. The detail of different stages are given in the figure legends.
Please, correct also the description in the text as not all the information that you write about the results is correct in relation to what is presented in the SF2.
Thank you. We corrected the text.
Line 193
To make it easier for the reader, please, write everywhere the same name for the stages analyzed, and in the same order – both in the text here and in the one describing the figure.
Thank you. We modified the text as suggested
Line 198
It rather suggests functions of these genes at various plant developmental stages and in different tissues, not necessarily different ones.
Thank you. We modified the text as suggested
Line 201-202
Since white does not mean “0” according to the color scale applied, the blue for MDH11 expression in roots may actually mean 0 (?)
Thank you. We modified the text as well as updated the figures as suggested.
Line 202-204
I believe to write here about intermediate expression is not correct. Maybe in relation to other expression it could be described as intermediate. Try to rewrite it.
Thank you. We modified the text as suggested
Line 215
“in the middle” – I believe it is not needed here.
Thank you. We modified the text as suggested
Line 136-148
Either in the section Materials and Methods or here, describe in more details the treatments.
Please, correct the description for the results presented in the figure 5 as not all the information that you write about these results is true. There is quite substantial number of mistakes. To avoid such mistakes, I suggest to not write about the results in a too general way.
Thank you. We modified the text.
List the 7 genes that you write about,
Thank you, We modified the text.
Please, refer to the Recovery phase in the figure 5. What does it mean? Try to describe it more and explain these results as well.
Thank you, We modified the text and added the information on the recovery phase.
Line 153-155
Please, describe all abbreviations used in the figure. Moreover, I strongly suggest to make the scale more uniform, e.g. from -1 to 3 in all figures, and for “-“ results use the blue color, for “0” white, and for the “+” red. At this point it is quite confusing. I think some confusion in describing these results may results from this not uniform color code.
Thank you, We modified the figure according to your suggestions.
I like the improvements in the figure. Now it seems more clear. However, still, “0” as white would make it much better.
Please, describe all abbreviations used in the figure.
Thank you. We modified figures and text as suggested
Line 222-224
Actually, this is not correct. It seems like the expression of these four genes is relatively high independent of stress application as you see that it is high both at “0” time-points/in control samples and for treated ones. For MDH2 or MDH11 it even seems to drop along with the time of treatment. That has to be very well though through and written correctly. Such interpretation changes also slightly the idea of your work. You have to be very careful when writing the abstract, and conclusions.
Thank you. We modified the text in the abstract and conclusion as suggested
Line 225-226
MDH1 – probably yes, but not for MDH7 (it is probably below 0 but does not decrease along with the stress). Keep also in mind that for salt stress you have timepoints both for the control and treated samples. That also have to be correctly interpreted.
Thank you. We modified the text as suggested
Line 227-229
The interpretation is not fully correct.
Thank you. We modified the text as suggested
Line 229-230
Repetition of what was written earlier.
Thank you. We modified the text as suggested
Not “from RNA-sequencing” but e.g. “based on RNA-sequencing data” or “using RNA-sequencing data”
Thank you, We modified the text.
Line 159
Not “by exposing tomato plants to salt stress” but “on plants exposed to salt stress”
Thank you, We modified the text.
Line 257
Maybe not “interval” but “0, 3, and 6 hours after treatment” (here and in figure description).
Thank you. We modified the text as suggested
Line 174
Statistical analysis of difference significance is needed for the gene expression studies.
Thank you, We applied the T-test.
Line 273
How many biological replicates have you used?
Thank you. We used 3 biological replicates and modified the text.
Line 183-184
There is no information on residues in Figure 7. It would be actually good to add such information.
Thank you. The interacting residues and binding information are already given in table 2. In addition, it is a lot easier to compare the four proteins in table 2.
Line 280
Considering the changes in interpretation of the RNA-seq expression data I think “highly expressed in all the abiotic stresses” does not fit here.
Thank you. Yes, I agree and now We modified the text as suggested.
Line 195
Correct the result for oxaloacetate
Thank you. we corrected the text.
Line 194-195
“However, the NADH molecule showed the strongest binding with all the SlMDHs proteins” – not correct, for SlMDH2 – the same for NAD and NADH
Thank you. we corrected the text.
Line 204
“2, 4 and 3 hydrogen bonds” not “2, 4 and 2 hydrogen bonds”
Thank you. we corrected the text.
Line 219
What about NADP? How do you know which MDH has NAD+ or NADP+ cofactor?
According to the malate valve hypothesis, NADPH is generated in the chloroplast through the electron transport chain during the day but does not readily diffuse out of the chloroplast. NADP-MDH reduces oxaloacetate to malate by oxidizing NADPH to NADP, and the malate is shuttled to the cytosol in exchange for oxaloacetate via the dicarboxylate transporter. In the cytosol, malate is oxidized back to oxaloacetate by the cytosolic MDHs, regenerating a reducing equivalent in the form of NADH in that compartment. So when the MDH is located in chloroplast it has NADP+ cofactor, while cytosolic MDH has NAD+ cofactor.
I suggest to explain in the publication why you did not carry out these analyses with NADP+ for the MDH11, which likely localizes in chloroplasts.
Thank you. Actually, we followed this analysis according to this paper (2020 Aug 7;10(1):13372. doi: 10.1038/s41598-020-70202-y). we followed all the steps. As we are doing this analysis in plants (having chloroplast). it needs time to set a multiple test hypothesis and get the results. Thus, we are unable to show the results in this publication. However, we have considered it for future research and
Reference..?
(2020 Aug 7;10(1):13372. doi: 10.1038/s41598-020-70202-y).
It is known from kinetic studies that the malate to oxaloacetate reaction is an ordered reaction with NAD/NADH binding first, followed by the dicarboxylic acid substrate (Silverstein et al. 1969). It would mean that the docking for malate has to be done for the MDH molecule with the cofactor. Is it possible to do it?
This is possible but it will need a lot of time as this is a very detailed study and requires a lot of computational resources. However, this is not the center of the current study, but we thank the reviewer for a nice suggestion.
Please, describe the meaning for all abbreviations used in the table.
Thank you. we modified the text by adding the abbreviations for each.
Line 222-225
What are these ligands exactly with particular SlMDH? Please, add that information.
I suggest to write just “blue” as there are different blue colors for proteins in figure 7.
Thank you. we modified the text.
Line 228-237
Please, cite all the QTL studies that were used by you to mark QTLs in Figure 8. I also suggest to at least write their names while describing them in the text.
Thank you. we cited the references and modified the text.
Line 351
Please, correct names of QTLs so they are as in the figure.
Thank you. We modified the text as suggested
Line 337-358
It seems like more MDH genes colocalize with the chosen QTLs. It is then not clear while you still claim (including abstract) that especially MDH11 colocalizes with salt stress-related QTL. That has to be clearly explained. You should make clear statement which of the described characteristics are salt stress-related.
Based on relatively higher expression, colocalization and Chloroplast subcellular location we selected MDH11 as putative gene. As we know “in reaction to unfavorable environmental pressures, chloroplasts communicate their status with the nucleus through a process called retro-grade signaling, which helps regulate the nuclear stress response
Discussion
Line 254
“MDHs are very…” not “MDH is a very”
Thank you. we modified the text.
Further part of this sentence has to be also corrected (now line 367).
Thank you. We modified the text as suggested
Line 373
The sentence is missing a part.
Thank you. We modified the text as suggested
Line 378-380
That is not really true.
Thank you. We modified the text as suggested
Line 383
When protein name-no italics.
Thank you. We modified the text as suggested
Line 395
That is not your group I but III
Thank you. We modified the text as suggested
Line 414
Gene names – italics
Thank you. We modified the text as suggested
Line 418
I would not come to conclusion that each gene exhibits a distinct expression.
Thank you. We modified the text as suggested
Line 254-263
There are some unnecessary repetitions, and quite some grammatic mistakes.
Thank you. we modified the text.
Citation [24] is not about MDHs.
Thank you. we corrected this mistake.
Line 268
These genes are not localized in chloroplasts but their products.
Four groups not two.
Thank you. we removed the text as it was not fitting properly in the context of the paragraph here.
Line 270
I am already confused whether you analyzed the motifs using gene sequences or amino acid ones. Here you write genes, in lines 80-82 protein sequences, and then in Material and Methods (4.2) – genes. That has to be clarified.
We apologize for the inconvenience. We used the amino acid sequence for motifs, and we mentioned it in the text as well.
Line 274
You did not show or write such results. I suggest to transfer this information to the results part.
Thank you. we transfer this information in the text.
Line 296
Please, write names of the duplicated genes.
Thank you. we modified the text.
Line 302-303
There were no 3 groups for the RNA-seq data.
Thank you. we modified the text.
Line 304-307
You have to be mor specific. That does not apply to all the genes studied.
We apologize for the inconvenience. We removed the sentences it was misleading and did not properly present the facts.
Line 307-308
There is no previous information (and in Materials and Methods) that the expression was analyzed in roots. All this has to be more explained.
Thank you. we modified the text.
I suggest to mention that also in part Results.
Thank you. We modified the text as suggested
Line 310
“especially the SlMDH2, and SlMDH11 gene at 3hr” – it is not correct – 2 and 12.
Thank you. we corrected the text.
Line 425
The word “respectively” is missing. At this point, it is understood as both gene were highly expressed at both timepoints.
Thank you. We modified the text as suggested
Line 437
Please, take into account the proper interpretation of the data.
Thank you. We modified the text as suggested
Line 315-317
This information is not correct.
Thank you. we corrected the text.
Line 324-325
“Our docking results showed that binding affinity of SlMDH11 is similar between oxaloacetate and malate,” – it was similar for other analyzed SlMDHs as well
Thank you. The binding affinity was slightly higher for SlMDH2, SlMDH6 and SlMDH9 for oxaloacetate over the malate, but it was the same for MDH11.
In my opinion there are no reasons to somehow say that from all the identified and characterized MDHs in tomato, the 11th one is somehow more involved in salt stress responses. Thus, the title has to be also adjusted accordingly.
Thank you. We agree with the reviewer that it is a tough decision to call out for SlMDH11. However, among the four highly expressed (SlMDH2, SlMDH6, SlMDH11, and SlMDH9), only SlMDH11 was co-localized with QTLs. We change the title as recommended.
Material and Methods
Line 447
What kind of sequences – nucleotide, amino acid?
Thank you. We used amino acid sequences and modified the text as suggested
Line 451
In InterProScan I believe you used amino acid sequences not gene ones?
Thank you. Yes, its true. Now we modified the text as suggested.
Line 487
You already stated that motif analysis was carried out with amino acid sequences. Thus, it can not be written as “motif analysis of MDH genes”.
Thank you. Yes, its true. Now we modified the text as suggested.
Line 334
BLAST is a tool not a verb.
Thank you. we corrected the text.
Line 335
“With the help of ProtParam Tool,” – please, add citation.
Thank you. we elaborated the text.
Line 343
“motifs were selected as 10 and the other parameters used were default” – instead, I suggest to write “motifs were selected as 10 and the other parameters were used as default”
Thank you. we modified the text.
Line 346
Not “4.3 Phylogenetic analysis and sequence alignment” but “4.3 Sequence alignment and phylogenetic analysis”
Thank you. we modified the text.
Line 350
Which MEGA was used – 6 or 7? (somewhere before you wrote 6).
Thank you. we used MEGA6 and we also modified the text.
Line 350-351
Try to combine it with the first sentence in this part of the text.
Thank you. we modified the text.
Line 355
The name of the program is PlantCARE. Please, cite it.
Thank you. we cited as suggested.
Line 358-365
This part – “Synteny and collinearity analysis” – should be placed before “4.3 Phylogenetic analysis and sequence alignment”. Part 4.2 should go after the Phylogenetic one.
Thank you. we modified the text as suggested.
Line 367-377
Cite the databases. I believe more information on treatments has to be written.
Thank you. we modified the text.
Line 509-513
There are some repetitions.
Thank you. We modified the text as suggested.
Line 381
The tomato name has to be editd.
Thank you. we modified the text.
Line 382
Seeds were sown and plant were grown at
Thank you. we modified the text.
Line 384
How old were these seedlings?
Thank you. Two weeks seedlings were used for treatment and samples were harvested after 3 and 6 hours.
How old were these seedlings when transferred to the Hoagland solution?
Thank you. One week old seedlings.
Line 385
Please, write the composition of the solution.
Thank you. we modified the text.
Line 386-388
What was the volume of the original solution and then how much of the 250 mM solution was added?
Thank you. we modified the text.
You mean you added so much of the NaCl solution so the medium was eventually 250 mM NaCl (we do not write mmol/L)?
Thank you. Yes, its true. Now we modified the text as suggested.
Line 391
You do not show results for these timepoints.
We apologize for the inconvenience. We corrected the text.
Line 396
To the list in the ST2, please, add primers for actine.
Thank you. we added the primer.
Line 397
Please, add citation for the method used to calculate relative expression.
Thank you. We cited the method we used for expression analysis.
In the Results part you write that you applied t-test. I do not understand then why you write about ANOVA.
Thank you. It was mistakenly written. We modified the text as suggested.
Line 4
04
Citations for the sources
Thank you. We cited the references.
References
Please, remember about writing Latin names of species in italics.
- I guess it is better to write as authors “Sato S,…. and the other authors (till the number of authors allowed by IJMS). The “x” is I guess not needed.
Thank you. Actually, there were more authors in this publication’s we checked the other papers also and got the same result as we used.
- I believe there are some mistakes in the citation.
Thank you. We corrected the references
- The paper that you cite is just a review. I suggest then to rather cite the program with the website of use, according to the standards of IJMS. Otherwise, I believe the review should be cited including following information: “J. Chem. Inf. Model. 2010, 50, 11, 2053. “ (in a way adjusted to the requirements of IJMS).
Thank you. We corrected the references

Round 3
Reviewer 1 Report
Dear All,
The manuscript has been improved. However, I do still have some comments, which you can find in the attached file. I also suggest to read the text carefully and check it with the data presented in figures. Again, the interpretation of the expression data is the major drawback of the manuscript.

Author Response
Dear Reviewer,
We appreciate the reviewer for excellent suggestions to improve the quality of the manuscript. We carefully improved the manuscript in the light of reviewer suggestions. All the answers for 3rd revision were written in a green bold color in the attachment.
